# The external validity of machine learning-based prediction scores from hematological parameters of COVID-19: A study using hospital records from Brazil, Italy, and Western Europe

Ali Safdari[1], Chanda Sai Keshav[1], Deepanshu Mody[1], Kshitij Verma[1], Utsav Kaushal[1], Vaadeendra Kumar Burra[1], Sibnath Ray[2], Debashree Bandyopadhyay[1] *

**1** Department of Biological Sciences, Birla Institute of Technology and Science, Pilani, Hyderabad Campus, Hyderabad, Telangana, India, **2** Gencrest Private Limited, 301-302, B-Wing, Corporate Center, Mumbai, India

☯ These authors contributed equally to this work.
* banerjee.debi@hyderabad.bits-pilani.ac.in

## Abstract

The unprecedented worldwide pandemic caused by COVID-19 has motivated several research groups to develop machine-learning based approaches that aim to automate the diagnosis or screening of COVID-19, in large-scale. The gold standard for COVID-19 detection, quantitative-Real-Time-Polymerase-Chain-Reaction (qRT-PCR), is expensive and time-consuming. Alternatively, haematology-based detections were fast and near-accurate, although those were less explored. The external-validity of the haematology-based COVID-19-predictions on diverse populations are yet to be fully investigated. Here we report external-validity of machine learning-based prediction scores from haematological parameters recorded in different hospitals of Brazil, Italy, and Western Europe (raw sample size, 195554). The XGBoost classifier performed consistently better (out of seven ML classifiers) on all the datasets. The working models include a set of either four or fourteen haematological parameters. The internal performances of the XGBoost models (AUC scores range from 84% to 97%) were superior to ML models reported in the literature for some of these datasets (AUC scores range from 84% to 87%). The meta-validation on the external performances revealed the reliability of the performance (AUC score 86%) along with good accuracy of the probabilistic prediction (Brier score 14%), particularly when the model was trained and tested on fourteen haematological parameters from the same country (Brazil). The external performance was reduced when the model was trained on datasets from Italy and tested on Brazil (AUC score 69%) and Western Europe (AUC score 65%); presumably affected by factors, like, ethnicity, phenotype, immunity, reference ranges, across the populations. The state-of-the-art in the present study is the development of a COVID-19 prediction tool that is reliable and parsimonious, using a fewer number of hematological features, in comparison to the earlier study with meta-validation, based on sufficient sample size

**Data Availability Statement:** All relevant data are shared within the manuscript, Supporting information files (S1_Table.docx), and in github repository (https://github.com/debashreebanerjee/CoviPred).

**Funding:** DB received the research grant award funded by Department of Science and Technology, Mathematical Research Impact Centric Support (MATRICS) (DST-MATRICS) COVID-19 special call, Government of India, grant No: MSC/2020/000498. URL: https://dst.gov.in/ AS received Junior and Senior Research Fellowships, Award Number: 09/1026(0033)/2020-EMR-I from Council of Scienctific and Industrial Research, Government of India. Funders has no role in the study design, data collection and analysis, decision to publish, or preparation of the manuscript.

**Competing interests:** The authors have declared that no competing interests exist.

**Abbreviations:** ALT, Alanine transferase; AUC, Area under curve; CBC, Complete blood count; COVID-19, Coronavirus Disease 2019; CRP, C-Reactive Protein; d-CFL, Ratio of (C-reactive protein times Ferritin) divided by White blood cell count; d-CIT, Product of C-reactive protein times International normalised ratio times Troponin; d-CT, Product of C-reactive protein times Troponin; d-CWL, Ratio of C-reactive protein divided by (White blood cell count times Lymphocyte count); d-PPT, Product of Platelet count times Procalcitonin times Troponin; ESR, Erythrocyte sedimentation rate; FAPESP, Fundação de Amparo à Pesquisa do Estado de São Paulo (São Paulo State Research Support Foundation); FN, False negatives; FP, False positives; FPR, False Positive Rate; GGT, Gamma-glutamyl transferase; ICU, Intensive care unit; INR, International normalised ratio; LDH, Lactate dehydrogenase; MCH, Mean corpuscular haemoglobin; MCHC, Mean corpuscular; hemoglobin concentration; MCV, Mean corpuscular volume; ML, Machine Learning; MPV, Mean platelet volume; OSR, Ospedale San Raffaele (San Raffaele Hospital); PT, Prothrombin Time; qRT-PCR, quantitative-Real-Time-Polymerase-Chain-Reaction; RAT, Rapid-antigen test; RBC, Red blood cells; RDW, Red blood cell distribution width; ROC, Receiver operator curve; SARS-CoV-2, Severe acute respiratory syndrome coronavirus 2; TN, True negatives; TP, True positives; TPR, True Positive Rate; W.E., Western Europe; WBC, White blood cell count; XGBoost, Extreme Gradient Boosting.

(n = 195554). Thus, current models can be applied at other demographic locations, preferably, with prior training of the model on the same population. Availability: https://covipred.bits-hyderabad.ac.in/home; https://github.com/debashreebanerjee/CoviPred.

## Introduction

The COVID-19 infection has posed the deadliest threat to the health of the human population in the 21$^{st}$ century. Likely, the danger is far from over concerning the emerging variants of COVID-19, such as alpha (B.1.1.7), beta (B.1.351), gamma (P.1), delta (B.1.617.2), lambda (C.37), and omicron (B.1.1.529) [1], along with other frequently mutating respiratory diseases, like, influenza virus A (H1N1) [2]. The most common clinical feature of COVID-19 is pneumonia with fever, cough, fatigue, headache, diarrhoea, hypoxia, and dyspnoea. The latest variant, omicron, has some common symptoms with the earlier SARS-COV-2 strains, although with lesser severity due to mild infection in the lower respiratory tract and reduced probability of hospitalization [1,3]. In the case of mild COVID-19 infection, either no (asymptotic) or only mild pneumonia is observed. In moderate infection, dyspnoea, hypoxia, and lung injury may occur. In severe infection, respiratory failure to multi-organ failure occurs. In brief, severe cases of COVID-19 can lead to a systemic infection affecting almost all of the major organ systems. Due to the nature of the disease, timely detection of COVID-19 is of utmost importance. Hence, detection techniques play a pivotal role in its diagnosis. There are two major types of COVID-19 tests: a) molecular tests (qRT-PCR tests) and b) rapid antigen tests. There is a less common antibody-based detection technique. Molecular qRT-PCR tests are considered as the gold standard for COVID-19 detection those detect the load of viral RNA in a patient. The sensitivity value (low false negative rate) for qRT-PCR ranges from 88–96% [4,5]. Although qRT-PCR tests are considered gold-standard, it has several limitations, like manual errors during sample (nasal and oral swab) collection, operational errors, etc. [6]. Moreover, the time required for the experiment and availability of the detection kits at a mass level becomes difficult in a vast population with a large number of infections. The test is also costly for low-income groups. The rapid-antigen test (RAT) is an alternate to qRT-PCR that detects the load of viral protein in an individual, that is much faster to qRT-PCR (on average 15 minutes only). RAT results have high specificity range 98%-99% (low false positives), but low sensitivity value 70%-72%, [7–9] This is because, RATs are more sensitive in the symptomatic and transmissive stages of disease when the viral load is higher [7–10]. The advantage of RAT is it can be done at mass-scale [11], and provide point-of-care by self-testing. Thus, in addition to their advantages, both qRT-PCR and RAT techniques have their own limitations. Development of a rapid, accurate and low-cost detection protocol can circumvent the short-coming of both the methods and supplement initial screening in large-scale and for low-income populations, particularly in a country like India, with the second-largest population in the world.

Literature reports indicate availability of several hematological bio-markers in COVID-19 patients, thus making those as potential candidates to develop alternate protocol. Patients of COVID-19 exhibit a wide range of hematologic abnormalities that changes with disease progression, severity, and mortality [12]. For example, the white blood cells sense [13]. and respond to microbial threats [14]. Similarly, blood platelet expression and platelet counts are altered in COVID-19 patients [15–17]. Platelet hyperactivity was demonstrated as one of the unique features of COVID-19 infection [18]. Abnormal levels of CRP, D-dimer, Procalcitonin, Troponin values were observed in the deceased. The most effective mortality biomarkers

identified were ESR, INR, PT, CRP, D-dimer and Ferritin. Neutrophilia, leukocytosis and erythrocytopenia were identified as mortality risk predictors [13]. As per the earlier reports, high d-CWL and d-CFL values largely confirmed the Covid-19 diagnosis. d-CIT, d-CT, d-PPT biomarkers were efficient in prognosis of COVID-19 disease [19]. White blood cell counts (WBC), lymphocyte counts, C-reactive proteins (CRP), D-dimer were used in prognosis and diagnosis of COVID-19 [20]. Hence, a complete blood count (CBC) could serve as a biomarker for COVID-19. Screening the COVID-19 infection in terms of CBC has been attempted by various research groups worldwide, [21–26].

Machine learning approaches were reported in literature for COVID-19 disease prediction, based on incident moments [27], SEIR models [28] etc. Some of the research groups used machine learning (ML) approaches to exploit the haematological parameters for prognosis, diagnosis and risk factor predictions [29–35]. From a specific population for disease prediction; the Area Under Curve (AUC) performance ranges from 84% to 87% in those models. So far, only handful of reports are available to test the applicability of the haematology-based ML models across different ethnicity and populations which has not been explored in previous studies [36]. The combination of haematological parameters varies with ethnicity in non-COVID individuals, for example, mean corpuscular (fL) and white blood cell counts ($10^9$/L) differ among African-American and whites [37]. A study conducted at a hospital of San Francisco between April 2017 and January 2018, showed that the reference intervals of neutrophil, lymphocyte and eosinophil counts; hemoglobin, mean corpuscular volume (MCV), mean corpuscular hemoglobin (MCH) and mean corpuscular hemoglobin concentration (MCHC) were significantly different ($p < 0.05$) across four racial/ethnic groups, namely, Asian, Black, Hispanic and White [38]. Various blood biomarkers (WBC, CRP, eosinophil, monocytes, platelets etc.) varied since inception of COVID-19 in 2020, world-wide, till date [17]. A set of blood biomarkers vary across severe (ICU patients) and non-severe (non-ICU) COVID-19 patients [39,40]. A study conducted in China in January 2020, showed that patients admitted with COVID-19 were reported with lymphocytopenia (83.2% of the patients), thrombocytopenia (36.2%), and leukopenia (33.7%) [41]. Similar study conducted in South Africa (March to June 2020) showed that decrease in median lymphocyte count and rise in d-dimer have no significant association with outcome [26]. Another study conducted in India (August 2020 to January 2021) noted severe elevation of D-dimer level in some patients, along with haemoglobin, Red Blood Cells (RBC) count, haematocrit, neutrophil and lymphocyte [42]. All these observations together suggested that race/ethnicity-specific variations occur in CBC parameters, both for reference intervals and COVID-19 patients. Furthermore, a recent study from Iran showed that hematological parameters, varied even within the same ethnicity, across different COVID-19 pandemic waves [43]. The authors showed that MCV and RDW-CV have increased during first wave, whereas, lymphocyte count, MCHC, PLT count, and RDW-SD have highest increase during second wave and so on. However, alteration in some of the blood parameters leading to lymphocytopenia [41,44] leucopenia, and thrombocytopenia [45–47], are more or less common due to COVID-19.

Based on the above observations, we hypothesized that ML model developed on haematological parameters would yield the best (COVID-19) probabilistic predictions when trained and (externally) validated on the same populations. The hypothesis, to some extent, was supported by the literature report where the authors showed external validation of the ML model trained on CBC parameters from Italy (training dataset) and externally validated on three other Italian datasets produced high sensitivity values (ranging from 85% to 91%). In contrast, low sensitivity values were produced when the same model was externally validated on three Brazilian datasets (ranging from 29% to 37%) [36]. In the current study, we aim to optimize the features in ML models those can be at least acceptable (in terms of meta-validation results) across the

populations. eXtreme Gradient Boost (XGBoost) model was benchmarked as the best-performing model across the datasets compared to published literature. The state-of-the-art of the current work is development of a COVID-19 prediction tool that is reliable and parsimonious, based on sufficient sample size (n = 195554).

## Method

### Description of clinical datasets for training, validation, and prediction

There are three major datasets curated from publicly available hospital sources (https://www.kaggle.com/einsteindata4u/covid19; https://zenodo.org/record/4081318#.X4RWqdD7TIU; https://repositoriodatasharingfapesp.uspdigital.usp.br/).

**Dataset 1.** Dataset-1 was generated based on anonymized patient data publicly available from Hospital Israelita Albert Einstein, in São Paulo, Brazil https://www.kaggle.com/einsteindata4u/covid19. The data were recorded from February 26[th], 2020, to March 23[rd], 2020. The patients hospitalized under i) regular ward, ii) semi-intensive care unit and iii) intensive care unit were included in this study. The cases and controls for this dataset include the patients whose samples were collected to perform the SARS-CoV-2 RT-PCR and additional laboratory tests during a visit to the hospital.

The initial data set consisted of 558 positive and 5086 negative cases of COVID-19. This dataset was processed to minimize the null-value columns and eliminate the negative instances with many null values. The value ($x_i$) in each cell was pre-normalized (at the source) to a mean value ($\mu$) of zero and a unit standard deviation ($\sigma$); this was termed as 'normalized count'; $x_i' = (x_i-\mu)/\sigma$. The same normalization scheme has been used throughout the subsequent datasets. The columns with null values appearing more than 90% were dropped. The initial data size was already small hence, usage of lower cutoff values reduced the dataset beyond its usability for model training. The records (rows) showing positive results were retained by default, and the negative records were maintained only with more than 10% non-null entries.

Thus, the negative data size was dropped to 1446, enriching the relative size of the positive data; the negative to positive sample size ratio reduced to 2.59, four times less than that in the published model (11.51) [21]. This newly processed dataset, enriched with positive results, was termed as dataset 1. In total, dataset 1 contains thirty-seven features and 2004 records, 558 positives and 1446 negatives (Table 1). Here 'features' refer to x-parameters used to train the model; the definition excludes the y-parameter, SARS-COV2 results (positive or negative). This definition is consistently used in the subsequent datasets. These thirty-seven features were categorized into four classes, namely, i) age, ii) severity of the infection, iii) hematological features, and iv) co-morbidities (S1 Table in S1 Appendix). The hematology analyser used therein was not known. We have further processed dataset 1 by dropping the co-morbidities. Thus, the total number of features were reduced to 18 and the number of records was 602, with 83 positives and 519 negatives.

**Table 1. Statistics of the datasets.**

| Dataset | No. of entries (P+N) | No. positive cases (P) | No. of negative cases (N) | Default scale_posweight (= N/P) | No. of features used |
|---------|----------------------|------------------------|---------------------------|--------------------------------|----------------------|
| 1(i)    | 602                  | 83                     | 519                       | 6.25                           | 18                   |
| 2 (ii)  | 1388                 | 765                    | 623                       | 0.81                           | 31                   |
| 3 (iii) | 6488                 | 1955                   | 4533                      | 2.32                           | 27                   |

I. https://www.kaggle.com/einsteindata4u/covid19.

II. https://zenodo.org/record/4081318#.X4RWqdD7TIU.

III. https://repositoriodatasharingfapesp.uspdigital.usp.br/.

**Dataset 2.**   This dataset was obtained from San Raffaele Hospital (OSR), Italy [23]. Inclusion criteria was patients admitted to the emergency department of the hospital from February 19, 2020, to May 31, 2020. Patients were excluded those who have potentially confounding pathologies and other sources of bias, such as insufficient data availability and admitted between February and April, 2020. All patients admitted during May 2020, were included. In the original OSR dataset, there were 1736 entries with a total of 72 features, among those, 36 were haematological features. Fifty-two percent of the patients were COVID-19 positive, as determined by RT-PCR tests on nasopharyngeal swabs.

These 1736 entries were processed such that all rows (records) with more than 66% null values were dropped. The null values were dropped based on trial and error method, using other (higher or lower) values the dataset became either too small or too sparse. In case of dataset 1, the initial size was already small hence, usage of 66% cut off would make the dataset not fit for ML model training. The processed dataset contained 1388 records, 765 positives, and 623 negatives (Table 1). This dataset includes 31 features: age, sex, a feature for suspicion (representing subjective analysis of the patient by a physician), and 28 haematological parameters (Fig 1). Haematology analyser used was Sysmex XE 2100. The ratio of negative to positive records was 0.81, indicating greater number of positives than negatives (Table 1).

**Dataset 3.**   Dataset 3 was obtained from the Covid Data Sharing initiative created by a consortium led by FAPESP (Sao Paulo Research Foundation) and USP (University of Sao Paulo, Brazil). The data originated from three prominent private hospitals in Sao Paulo, Brazil—Fleury Institute, Sírio-Libanês Hospital, and Albert Einstein Hospital, from November 1st, 2019, to July 1st, 2020 (https://repositoriodatasharingfapesp.uspdigital.usp.br/). The data was anonymized from patients tested for COVID-19 (serology or RT-PCR). The haematology analyser used therein was not known.

The raw data obtained from the data sharing initiative had multiple rows (records) corresponding to individual patients containing different clinical features ("long-form" of the dataset). The "long form" of the dataset was converted, using an in-house Python code, to the "wide form," where one row corresponds to all the clinical features of a patient. The "wide form" of the dataset has 189227 records and 454 features. These 454 features were common, as there were duplicates in the column headers (due to different reference ranges) for some features. After deduplication, the feature number was reduced to 104. The non-duplicated features were further filtered by excluding the following conditions, i) no qRT-PCR results available, ii) all the rows with more than 66% null values (similar to dataset 2). Twenty-seven hematological indices (features) were identified based on the above cutoff (Fig 1). The final dataset size was 6488; 4533 negatives and 1955 positives.

### Description of the clinical dataset for blind prediction

**Western European dataset.**   This dataset was obtained from several Western European hospitals, as reported elsewhere [25]. The dataset includes the patients from the first day of hospitalization to nearly five weeks [25]. This published data was in the form of twenty separate tables that we merged into a single file comprising 2587 entries and thirty-seven features. According to the source authors [25], there are two stages of the disease, a) early stage, from day zero through three (total of four days), and b) advanced stage, comprising all the subsequent days. This blind prediction dataset includes only four hematological parameters consistent with sub-dataset 2-four-features.

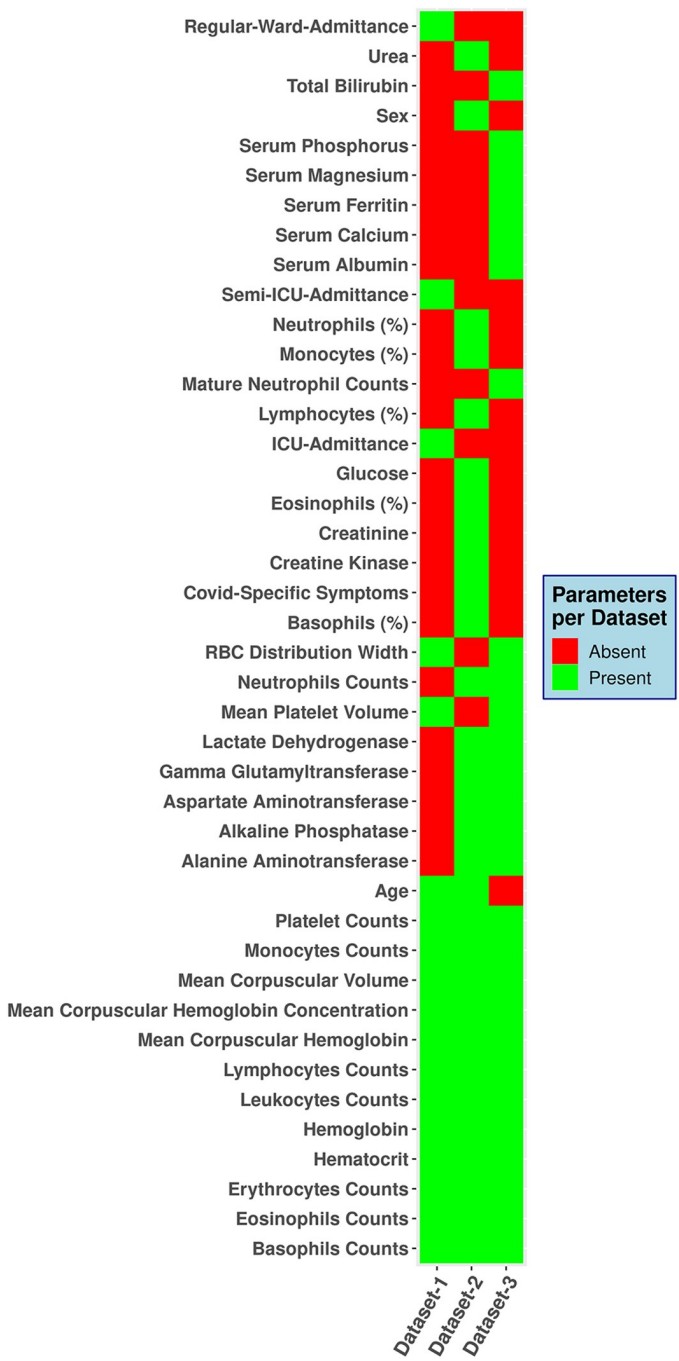

**Fig 1. Haematological features used in different datasets.**

## Machine Learning (ML) approaches

The machine learning (ML) algorithms were implemented in Python (3.7.13) using the following libraries, Numpy (1.21.6), Pandas (1.3.5), XGBoost (0.90), Scikit-learn (1.0.2), Seaborn (0.11.2), Matplotlib (3.2.2) and Pickle 4.0 libraries.

*Different algorithms*:

### Extreme Gradient Boost (XGBoost)

The algorithm primarily employed was the Extreme Gradient Boost (XGBoost) classifier that implements gradient-boosted decision trees (with enhanced speed and performance) and trains a class-weighted (or cost-sensitive) version of imbalanced classification [48]. XGBoost, a ternary classifier, considers null entries as one of the classes that handle the null-entry values.

Other classifiers tested on these datasets were logistic regression, Fischer linear discriminant Naïve Bayes, SVM, random forest, and K-Nearest Neighbor (KNN).

### Logistic regression

Logistic regression predicts the output of a categorical dependent variable by fitting an "S" shaped logistic function that indicates two maximum values, 0 or 1 [49].

### Fischer linear discriminant

Fischer linear discriminant classifier maximizes the separation between the projected class means and minimizes the class overlap leading to well-separated classes [50].

### Naive Bayes

Naive Bayes is a classification technique based on the Bayes theorem with an assumption of independent predictors; a particular feature is independent of another feature in a class [51].

### Support Vector Machine (SVM)

The SVM algorithm aims to create the best line or decision boundary to segregate n-dimensional space into classes to accommodate a new data point. The best decision boundary, a hyperplane, is made based on the extreme points (vectors) [52].

### Random forest

Random forest is a concept of ensemble learning–a combination of multiple classifiers to solve a complex problem and improve the model performance. As the name suggests, Random Forest contains several decision trees on various subsets of the given dataset and takes the average to improve the predictive accuracy of that dataset [53].

### K Nearest Neighbour (KNN)

KNN algorithm stores all the available data and classifies a new data point based on the similarity by placing a new data point in the nearest category. Thus, new data belongs to an appropriate class [54].

### Model training criteria

The proportion between training and testing sets is 90:10. 10-fold cross-validation with random split was performed on all the datasets.

**Hyper-parameter used in XGBoost classifier.**   To normalize the imbalance in the number of negative and positive data points in the XGBoost classifier, hyper-parameter–"*scale_pos_weight*" https://xgboost.readthedocs.io/en/stable/parameter.html#parameters-for-tree-booster, was introduced. The *scale_pos_weight* value was used to scale the gradient for the positive class. For example, the "*scale_pos_weight*" = 2 indicates twice the weight of the positive class compared to the negative class. It also overcorrects the misclassification of the positive class. The loss curve (optimized to get a better model) will be affected differently in case of

positive and negative entry misclassification. However, large scale_pos_weight can help the model achieve better performance for the positive class prediction (overfitting the positive class) at the cost of worse performance on the negative or both classes. Hence, we have consistently considered the default *scale-pos-weight* (the ratio of numbers of negative to positive entries) throughout this report.

**Imputation for other ML models.** Unlike XGBoost, most ML algorithms cannot handle null values, thus requiring data imputation. We imputed missing values through the IterativeImputer module in the ScKit-learn package (https://scikit-learn.org/stable/modules/impute.html#multivariate-feature-imputation), which imputes values for null data points for each feature iteratively. It does so by fitting a regressor to the other feature columns (X-parameter) for records with known values of the target feature (y-parameter) and then predicts missing values of the target feature. Chi-square test was performed between the imputed and non-imputed dataset which returns zero values, in almost all the cases (S2 Table in S1 Appendix). The null hypothesis tested in Chi-squared test was that two populations were significantly different (https://docs.scipy.org/doc/scipy/reference/generated/scipy.stats.chi2_contingency.html). Chi-squared test returning zero values indicated that the null hypothesis was wrong; hence, both the populations behave similarly.

**Performance metrics.** *Internal Evaluation.* Four metrics were used for internal evaluation of the models, namely, sensitivity, specificity, accuracy and AUC scores. The performance metrics were defined by true positives (TP), true negatives (TN), false positives (FP), and false negatives (FN) (Eqs 1–3). False negatives, in terms of diagnosis, are the cases where Covid-19 positive patients would be classified as negative and would be possibly allowed to go home. Thus, these would be more harmful than false positive cases–healthy individuals predicted as COVID-19 positive. In screening task, accuracy is defined as success in identifying patients and healthy individuals in total.

$$Accuracy = \frac{(True\ Positive + True\ Negative)}{(True\ Positive + True\ Negtive + False\ Positive + False\ Negative)} \quad (1)$$

$$Sensitivity = \frac{True\ Positive}{True\ Positive + False\ Negative} \quad (2)$$

$$Specificity = \frac{True\ Negative}{True\ Negative + False\ Positive} \quad (3)$$

For all the above-mentioned metrics, interval values were computed within 95% confidence limit using the formula (Eq 4).

$$interval = \frac{1.96 * \sqrt{(metrics * (1 - metrics))}}{sample} \quad (4)$$

the constant 1.96 stands for the number of standard deviations (1.96 for 95% confidence limit).

The fourth metric was the Area Under the ROC Curve (AUC). The AUC was computed from prediction scores using the roc_auc_score (https://scikit-learn.org/stable/modules/generated/sklearn.metrics.roc_auc_score.html) module of the sklearn—metrics library. A ROC curve (Receiver Operating Characteristic curve) plots the performance (True Positive Rate (TPR) versus False Positive Rate (FPR)) of a classification model at all classification thresholds. In terms of diagnosis, AUC determines positive cases as actual positive.

TPR is synonymous with sensitivity, also known as recall. FPR is FP/(FP + TN). AUC measures the Area under ROC (as defined by TPR versus FPR) curve from (0,0) to (1,1) along the x-axis (FPR axis). AUC ranges from 0 to 1; 0 implies a 100% wrong model, and 1 indicates a 100% correct model.

*External Evaluation.* In addition to the above metrics, few more measures were considered for external evaluation, capable of handling both balanced and imbalanced dataset.

*Validation of the sample size for the external datasets (dataset cardinality).* Minimum Sample Size (MSS) was computed to validate the sample size of the external dataset following the method described in the literature [36,55].

*Measuring data similarity between the training and the external validation datasets.* The data similarities among the training and the external validation datasets were determined using Kolmogorov–Smirnov (KS) test [56]. KS test is non-parametric that determines whether given two sample datasets come from the same distribution.

*Handling data imbalance between training and test datasets.* In order to handle the potential imbalance in the target distribution, "Balanced Accuracy", "F- beta scores" were computed, using sklearn module in python (https://scikit-learn.org/stable/modules/generated/sklearn.metrics.balanced_accuracy_score.html). "Balanced Accuracy" is defined as the average of sensitivity and specificity (Eq 5). The best value is 1 and the worst value is 0 when adjusted = False. The F-beta score is robust scoring scheme for balanced and unbalanced datasets. F-Beta accounts "precision" and "recall" together and performs a weighted harmonic mean between these two (Eqs 6–8). A harmonic mean is the average computed by adding the reciprocal of individual values in a data set and normalizing it with the total number of datapoints. To note, "precision" computes the percentage of correct predictions for positive class and "recall" computes the percentage of correct predictions for positive class out of all possible positives. A smaller value of beta gives more weight to Recall, while a large value of beta gives lower weight to Recall. The value of the F-beta score lies between 0 to 1 (1 is the best value). In this study we used beta equals to two.

$$\text{balanced-accuracy} = \frac{1}{2}\left(\frac{TP}{TP + FN} + \frac{TN}{TN + FP}\right) \tag{5}$$

$$Recall = \frac{TP}{TP + FN} \tag{6}$$

$$Precision = \frac{TP}{TP + FN} \tag{7}$$

$$F\beta = \left(1 + \beta^2\right) * Precision * Recal / \left(\beta^2 * Precision + Rcall\right) \tag{8}$$

TP, TN, FP and FN represent True positive, true negative, false positive and false negative, respectively.

*Model Calibration using Brier Score*: Brier Score evaluates the accuracy of a probabilistic prediction. It is more like a cost function [57]. In case of a binary prediction, the score is defined as in Eq 9.

$$BS = \frac{1}{n}\sum_{i=1}^{n}\left(p_i - o_i\right)^2 \tag{9}$$

Where $p_i$ is the probability of occurrence of the event"i" and $o_i$ is the actual outcome (0 or 1) of the event.

*Model utility using standardized net benefit*: Standardized net benefit ($sNB_{pt}$) was computed using the formula given by Riley et. al [58], Eq 10.

$$sNBpt - (sensitivity \times \varphi) - (1 - specificity) \times (1 - \varphi) \times pt/(1 - pt) \qquad (10)$$

Where $\varphi$ represents observed outcome event proportion; for example, if there are 83 COVID-19 positive patients in total sample size of 602, $\varphi$ will be 0.14. $p_t$ represents probability threshold, that is generally considered as 0.5, where fifty percent of the population is positive and remaining is negative. However, this is mostly not the real case. Earlier reports included $p_t$ = 0.08 for the highest risk group [59].

*Meta-validation of the external performances*: Meta-validation procedure was adopted from the method described in the literature [36]. Here, performances of the ML models were assessed in two dimensions, i) dataset similarity (between training dataset and external validation dataset), measured using KS test, described above and ii) dataset cardinality, measured in terms of minimum sample size (MSS), described above. The performance was evaluated in terms of discrimination (Balanced accuracy), utility (Standardized net benefit) and calibration (Brier score). Two sets of training-external validation datasets were used for meta-validation, in this study.

## Results and discussion

### Correlation between features and SARS-COV-2 results in different clinical datasets

Three independent clinical datasets (dataset 1, dataset 2, and dataset 3) were curated and processed from hospitals in Brazil and Italy. The point-biserial correlations (positive or negative) and p-values were computed between the features and the SARS-COV-2 results (Fig 2).

For dataset-1, following features were correlated (p-value <0.05) to SARS-COV-2 results, namely, age, regular ward admittance, IC admittance, hemoglobin, hematocrit, platelets, MPV, leukocytes, eosinophils and monocytes. Among the CBC parameters, monocytes, hemoglobin, hematocrit, RBC and MPV have shown significant increase in their values in SARS-COV-2 patients (positive point biserial correlation). The remaining parameters decreased during infection (negative point biserial correlation) (Fig 2). Careful observation revealed that in the case of non-admitted patients, the increase in monocyte is maximum, suggesting that innate immunity is handling the infection. On the other hand, platelet volume (MPV) increased, and platelet counts decreased in the case of regular ward patients, clearly indicating the increase in platelet size. Thus, the immune system will be affected, and the number of immune cells will decrease, justifying the negative correlation of eosinophil, leukocytes, and platelet count with SARS-COV-2 disease. The low platelet counts were accounted for severe COVID-19 patients, those were even down in non-survivors compared to the survivors [60]. The correlation coefficient values between SARS-COV-2 results and different features reported elsewhere were similar to these observations [23]. For dataset 2, twenty-six features were correlated (p-value <0.05) to SARS-COV-2 results (Fig 2). Following parameters showed positive point biserial correlations, namely, sex, aspartate amino transferase, alanine transferase, lactate dehydrogenase, hemoglobin, hematocrit and MCHC, i.e., these parameter values increased in COVID-19 patients. Negative point biserial correlations were observed in leukocytes, platelets, erythrocytes, eosinophils, basophils, neutrophils, lymphocytes, monocytes and basophils. For dataset 3, twenty-two features were correlated (p<0.05) to SARS-COV-2 results (Fig 2). Seven parameters, namely, serum ferritin, serum magnesium, MPV, lactate dehydrogenase, GGT, aspartate amino transferase and alanine transferase have showed positive point biserial correlations with SARS-COV-2 results. Whereas basophil, eosinophil, erythrocytes,

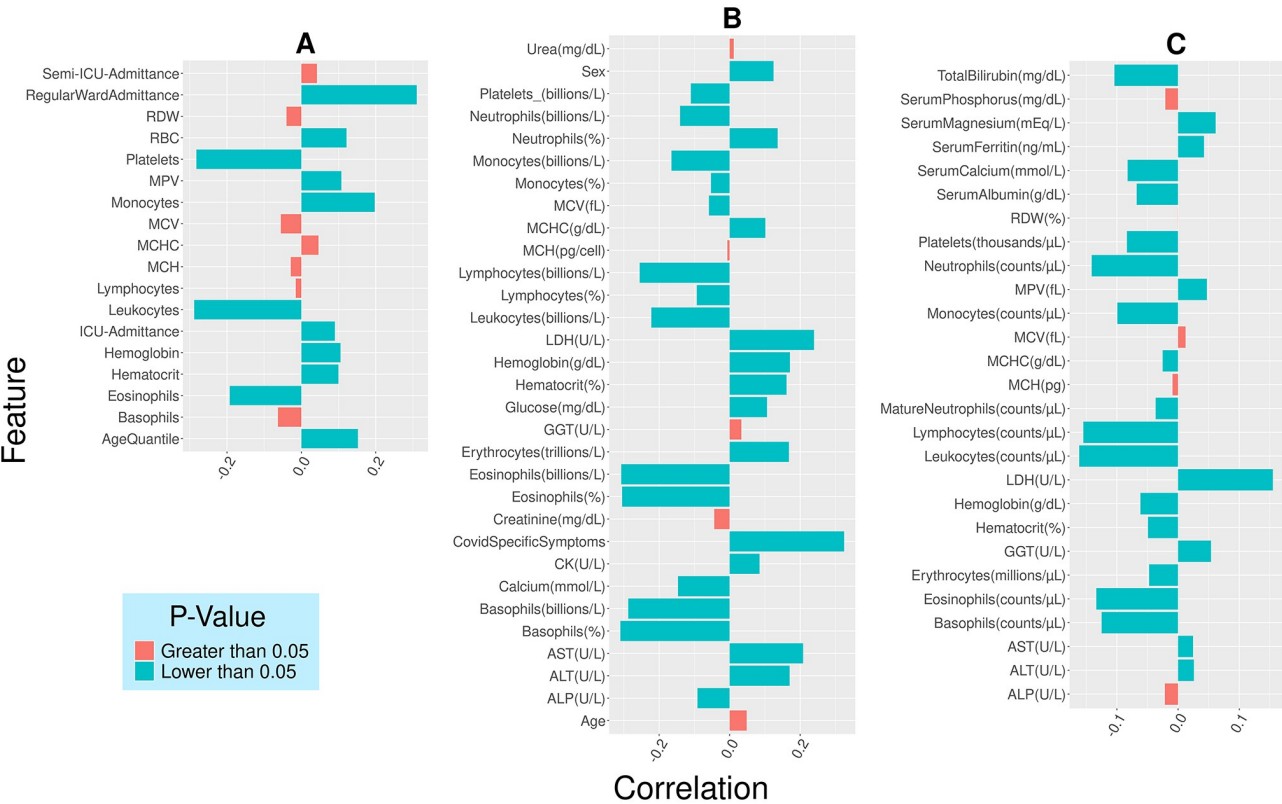

**Fig 2. Point biserial correlation coefficients between SARS-COV-2 results and individual features for a) dataset 1 b) dataset 2 and c) dataset 3.**
Parameters with p values < 0.05 are shown in blue, remaining values are in red.

hematocrits, hemoglobin, leukocytes, lymphocytes, neutrophils, monocytes, platelets, MCHC, serum albumin, serum calcium and total bilirubin have showed negative point biserial correlations with SARS-COV-2 results.

While comparing datasets, 1 to 3, few features–those correlated with SARS-COV-2 results (p<0.05) were observed common across the datasets. Those features were, hemoglobin, hematocrit, platelets, leukocytes, eosinophils and monocytes. These features either decrease (negative point biserial correlation) or increase (positive correlation) in SARS-COV-2 patients, in different degrees according to the disease severity. Platelets, leukocytes, and eosinophils consistently showed negative correlations with SARS-COV-2 results, across all the datasets. Monocytes, also showed negative correlations with SARS-COV-2 results, except in dataset 1. Other parameters, like, basophils, neutrophils, lymphocytes, showed negative correlations with SARS-COV-2 results, in both datasets 2 and 3. Hemoglobin and hematocrit showed positive correlations in dataset 2, whereas, negative correlations in dataset 3. The observations indicated the dependencies of certain hematological parameters on demographic populations. In summary, features correlated (p<0.05) with SARS-COV-2 results across all three datasets were, hemoglobin, hematocrit, platelets, leukocytes, eosinophils and monocytes. The features common across datasets 2 and 3 were, hemoglobin, platelets, leukocytes, eosinophils, monocytes, hematocrit, erythrocytes, lymphocytes, basophils, neutrophils, LDH, serum calcium, MCHC and ALT. To generalize further, we compared these observations with hematological parameters obtained from Indian populations, those correlated with D-dimer values [42]. Henceforth, this dataset will be mentioned as Indian dataset. The common features across

datasets 1, 2, 3 and Indian dataset, were platelets, eosinophils, monocytes and leukocytes. These four features were used earlier by Banerjee et.al., (on Brazilian dataset) for hematology-based ML model development [21]. Hence, the same set of features were used, here, for external validation of the models.

## Curation of working sub-datasets

We have curated three primary datasets from hospitals of Brazil and Italy. Based on the common hematological parameters across datasets, 1, 2, 3 and Indian dataset, a subset of four-features was developed from datasets 1 and 2 named as 1-four-features, and 2-four-features respectively. The 1-four-features sub-dataset contained 602 records, 83 positives, and 519 negatives. Thus, the negative-to-positive sample size ratio was 6.25. The 2-four-features sub-dataset contained 1736 records, 816 positives, and 920 negatives, with subsequent increase in the ratio of negative to positive records to 1.13 compared to 0.81 in dataset 2.

Apart from four-featured datasets, we have also considered standard full blood count [21,61]—hematocrit, haemoglobin, platelets, mean platelet volume (MPV), red blood cells (RBC), lymphocytes, mean corpuscular haemoglobin concentration (MCHC), leukocytes, basophils, neutrophils, mean corpuscular haemoglobin (MCH), eosinophils, mean corpuscular volume (MCV), monocytes and red blood cell distribution width (RBCDW). Most of these parameters were present across all the datasets; although, all might not have shown correlations ($p < 0.05$) with SARS-COV-2 results. This is termed as fourteen-featured dataset, developed from datasets 1 and 3 and named as, 1-fourteen-features and 3-fourteen-features. The 1-fourteen-features sub-dataset contains 602 records, 83 positives, and 519 negatives; the negative-to-positive sample size ratio was 6.25. The 3-fourteen-features sub-dataset included total 12105 records. The negative to positive ratio in this dataset was 0.356, that is, the dataset is mildly skewed towards positive results. Hence, a scale_pos_weight of 0.356 was used to treat the data imbalance.

These four different sub-datasets curated, here, were trained on seven different machine learning models. The summary of all the raw and processed datasets along with the four sub-datasets were represented as a scheme (Fig 3).

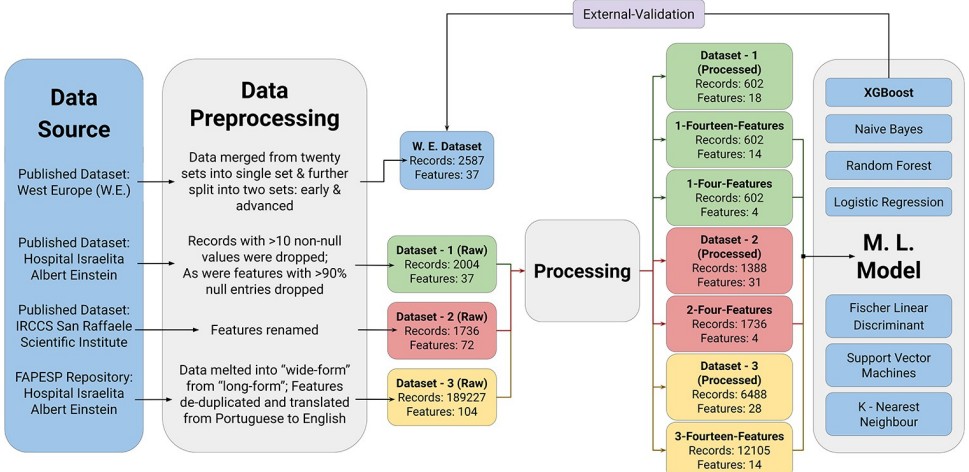

**Fig 3. Description of data sources used for training, testing and external validation of different ML-models based on haematological features for COVID-19 characterization.**

### Benchmarking of seven different ML models on working sub-datasets

The 10-fold cross validation studies on four sub-datasets showed that random splits for 10-fold validation did not affect the performance metrics (nominal changes observed in standard deviation values, ranging from 0.01 to 0.19) (S3 Table in S1 Appendix). The performances of different models on four sub-datasets were measured using the receiver operating characteristic (ROC) curves (Fig 4). Receiver Operating Characteristic curve plots the performance (True Positive Rate (TPR) versus False Positive Rate (FPR)) of a classification model at all classification thresholds. In terms of diagnosis, AUC curves determine positive cases as actual positive. For all the four sub-datasets, ROC curve showed the optimal performance for XGBoost model. XGBoost model, being a ternary classifier, can handle the class imbalance and perform in a better way, compared to other ML models. To note, for other ML model data imputation was performed, yet the results were the best for XGBoost model. XGBoost outperformed other models for all the four sub-datasets, in terms of ROC curves. When all the performance metrics, namely, accuracy, sensitivity, specificity, and AUC scores were considered together, XGBoost still performed the best (S4 Table in S1 Appendix) The elapsed computational time was comparable across different ML models except support vector machine which has taken slightly longer time (S5 Table in S1 Appendix). Based on the above standardization, XGBoost model was selected for subsequent studies.

Sub-dataset, 1-four-feature, has shown optimal performance (AUC score 0.94) in all four metrics. The sensitivity value was observed as 1.0, indicating 100% correct prediction of True Positive (TP) values, presumably, due to overcorrection of the TP values in a small dataset (n = 602) with a low population of positives (n = 83), leading to large *scale-pos-weight* of 6.25. As mentioned in the method section, large *scale-pos-weight* improves the performance of the positive class prediction at the cost of the negative class prediction. Models other than XGBoost showed low-sensitivity values for 1-four-feature and 1-fourteen-feature datasets (S4

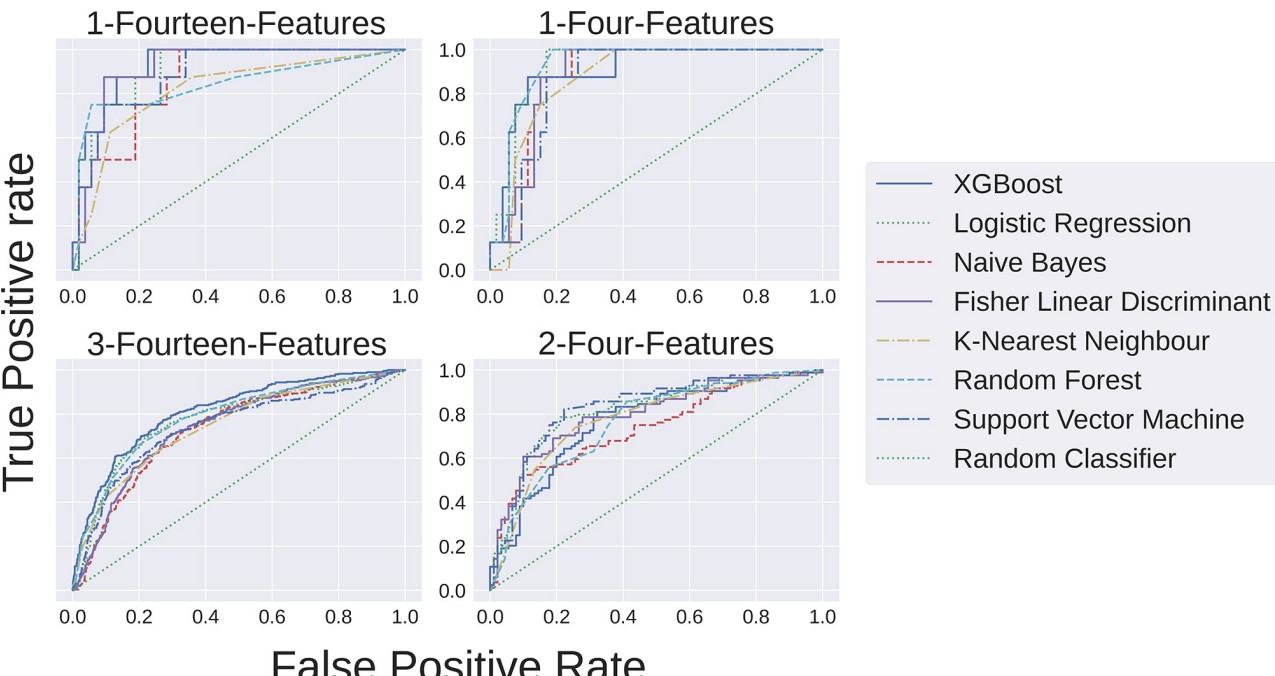

**Fig 4. Receiver Operating Characteristics Curves (ROC) across different ML models for four sub-datasets.**

**Table 2. Internal evaluation of the XGBoost model on different datasets and comparison with published datasets.** Interval computed within 95% confidence limit.

| Dataset | Sensitivity (±interval) | Specificity (±interval) | Accuracy (±interval) | AUC score | Published AUC score |
|---|---|---|---|---|---|
| 1-forteen-features | 0.750±0.11 | 0.887±0.08 | 0.869±0.08 | 0.922 | 0.87 (ref 21) |
| 1-four-features | 1.000±0.0 | 0.906±0.07 | 0.918±0.07 | 0.939 | 0.87 (ref 21) |
| 2-four-features | 0.845±0.05 | 0.733±0.07 | 0.787±0.06 | 0.842 | |
| 3-forteen features | 0.784±0.02 | 0.733±0.02 | 0.746±0.02 | 0.842 | |

Table in S1 Appendix). The low sensitivity values presumably attributed to the smaller data size and even a shallower positive population. Most likely, the XGBoost, being a ternary classifier, can more effectively handle the class imbalance than the imputations performed in other ML methods. However, the low sensitivity problem was absent in sub-dataset 2-four-feature, where the number of positives and negatives were equivalent. The performance of XGBoost model on 3-fourteen-feature sub-dataset was reasonable (Table 2) whereas, performances of the other models on the same sub-dataset was too low (S4 Table in S1 Appendix). Overall assessment indicated the general acceptability of XGBoost models over others.

## Comparison of internal performances of the XGBoost model with published reports

Purpose of internal validation of the XGBoost model was to evaluate the ability of the model to generalize on any given dataset.

The internal performances of the XGBoost model were compared with reported methods from the published literature (Fig 5). The XGBoost model developed on 1-four-feature and 1-fourteen-feature dataset showed AUC scores 0.92 and 0.94, respectively (Table 2). The sensitivity and specificity values obtained from XGBoost models on 1-fourteen-features sub-dataset were 0.75 and 0.89, respectively. Those from sub-dataset 1-four-features were 1.0 and 0.91 respectively (Table 2). These values were better than those reported by authors in [21] (sensitivity, 0.43 and specificity 0.91). The model by Banerjee et.al. [21] was developed on Kaggle dataset (same source, as described in this work, as dataset 1). Banerjee et. al. model was trained

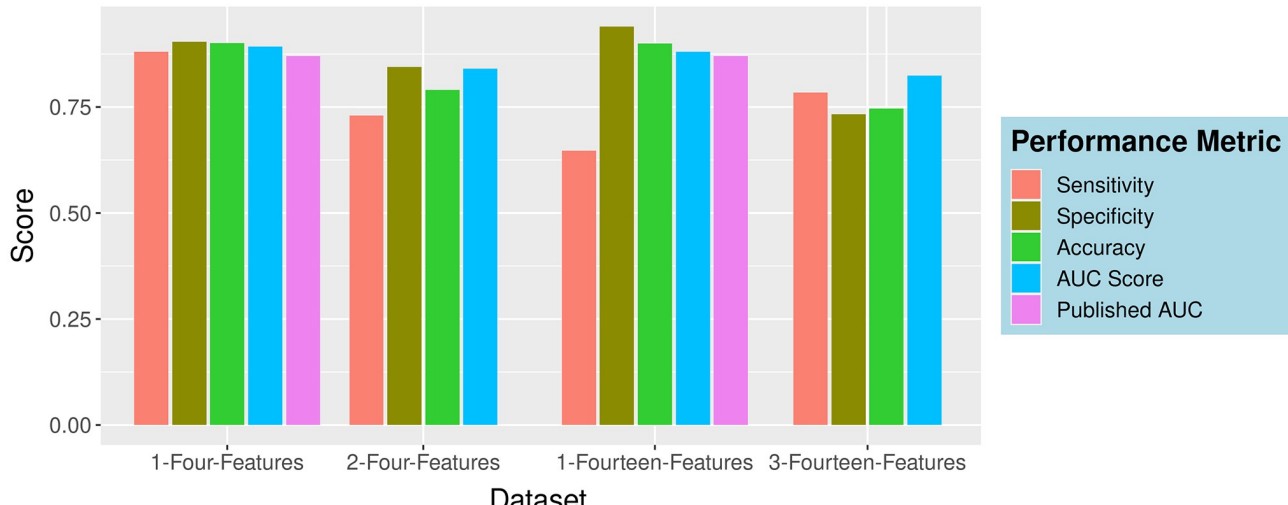

**Fig 5. Comparative performances of different sub-datasets trained on XGBoost model.** The datasets with published AUC scores [21] were compared.

on a dataset with skewed numbers of positive (n = 39) and negative cases (n = 598); the negative to positive ratio was much improved in the current work. Moreover, this model has not provided the information regarding the analytical instruments, analytical principle and the units of measurements, thus limiting the model in terms of replicability and generalizability. Using the same dataset [21], Avila et.al. [62] developed a Bayesian model with improved sensitivity and specificity values (76.7%). There were more reports in literature those described CBC-based ML models. Some of these models were developed on a very small dataset (n = 171) and limited time frame (from March 7th 2020 to March 19th 2020) [63]. Sensitivity and specificity reported from that work was 83% and 82%, respectively. Another research group [64] has developed a logistic regression model, trained on only 380 CBC data and reported high sensitivity (93%) but low specificity (43%) values. Other ML models, namely, Gradient Boosting (n = 3,356) [65] and K-nearest-neighbor and Random Forest (n = 1624) [23] were trained on large number of haematological and biochemical parameters and reported reasonable AUC scores, 0.85 and 0.78 respectively. Considering the overall raw sample size of the dataset used in this work (n = 195554) and the robustness of the XGBoost ternary classifier (ability to handle the class imbalance of the datasets, as evident from the results compared with other ML methods) (Tables 2 and S4 in S1 Appendix) this approach seems to be more effective in terms of reliability and generality.

## XGBoost models used for external evaluation across the populations

Purpose of the external evaluation was to test the sensitivity and specificity of the developed models on independent datasets and also to identify the potential suspect cases, by implementing XGBoost models.

As per the internal evaluation, the XGBoost model performed the best on the sub-dataset 1-four-feature. The performance of the XGBoost model on sub-dataset 1-fourteen-feature was comparable to that of sub-dataset 1-four-feature, with a slightly lower AUC Score (0.94 versus 0.92). Based on these scores, XGBoost models developed on sub-datasets 1-four-features and 1-fourteen-features tend to be attractive candidates for external evaluations. However, models developed on these datasets have some limitations, namely, overfitting problem due to small sample size. Moreover, the Kaggle dataset (the source of dataset 1) lacks the description of analytical instruments, analytical principles and measurement units; hence, models built on these training datasets have issues regarding reproducibility and generality [66]. Therefore, we selected two other XGBoost models with four and fourteen parameters obtained from two sub-datasets, sub-dataset 2-four-features (Italy), and sub-dataset 3-fourteen-features (Brazil), *albeit* with a slightly lowered AUC score of 0.842 in both cases. These two were the final working models (training dataset) for external evaluation.

## External evaluation of XGBoost models with four hematological parameters across Italian and Brazilian populations

External evaluation for the four-parameter model was performed on the test dataset 1-four-features from Brazil. Note that the training dataset 2-four-features was from Italy. The sensitivity was 0.81 with a low specificity value of 0.45. The dataset cardinality was evaluated using two metrics, AUC score and balanced accuracy; the score values were 0.69 and 0.63, respectively (Table 3a). The AUC score is sensitive to class imbalance, whereas balanced accuracy is better suited to handle imbalanced datasets and thus, is more reliable in this particular case. The data utility was computed in terms of standardized net benefit using the formulation reported earlier by Riley et al. [58]. The value was -2.6, at the prediction threshold value of 0.5. Considering the large class imbalance (ratio of positive cases to total sample size was 0.14), we used a

**Table 3. External evaluation of XGBoost algorithm based on a) 4- hematological features and b) 14-hematological features trained and tested across different datasets.** Interval computed within 95% confidence limit.

a)

| Training set/test set | Standardized net benefit[#] | Sensitivity (±interval) | Specificity (±interval) | Accuracy (±interval) | AUC Score | Balanced accuracy | Brier Score | F2-score |
|---|---|---|---|---|---|---|---|---|
| 2-four-features (Italian) / 1-four-features (Brazilian) | -0.08 (-2.61) | 0.81±0.002 | 0.45±0.002 | 0.50±0.002 | 0.69 | 0.63 | 0.28 | 0.49 |

b)

| Training set/test set | Standardized net benefit[#] | Sensitivity | Specificity | Accuracy | AUC Score | Balanced accuracy | Brier Score | F2-score |
|---|---|---|---|---|---|---|---|---|
| 3-fourteen-feature (Brazilian) / 1-fourteen-feature (Brazilian) | 0.33 (-0.072) | 0.55±0.002 | 0.90±0.001 | 0.85±0.001 | 0.86 | 0.73 | 0.14 | 0.80 |

[#] values computed at probability threshold of 0.14 and that within parenthesis was computed at probability threshold of 0.5.

different probability threshold of 0.15 that yielded a standardized net benefit score of -0.084. Both the values were worse than the "treat none" scenario—that is, no clinical decision administered to any patient. However, if we use a "treat all" situation, that is, all patients were treated (irrespective of COVID-19 positive or negative), the net benefit was -0.012 and 0.72, for probability threshold values of 0.15 ad 0.5, respectively. To note, standardized net benefit was computed earlier at different probability threshold values depending on the degree of risk factor [59].

## External evaluation of XGBoost models with fourteen hematological parameters within the Brazilian populations

The fourteen-parameter XGBoost model was trained on dataset 3-fourteen-features (n = 12105) and tested on dataset 1-fourteen-features (n = 602), both from Brazilian populations. However, the samples in these two datasets were from different time points; hence those can be considered independent data sources. The AUC score and balanced accuracy for this prediction were 0.86 and 0.73, respectively (Table 3b). These results were better than the performance for the four-feature XGBoost model across the populations. There could be multiple reasons for the better performance of the fourteen-feature model over the four-feature model, a) the larger size of the training dataset, b) training and prediction data obtained from the same demographic location, that is, Brazil, and c) combination of a greater number of features with a larger dataset, presumably, yields to a better result. The computed standardized net benefit [58], at the probability threshold of 0.5 was -0.07. Considering large imbalance in the COVID-19 positive cases in the external validation dataset (ratio of positive cases to total sample size was 0.14), the standardized net benefit was computed at a different probability threshold of 0.15 yielding the value of 0.33. Both the values at two different probability thresholds were fairly better than the "treat none" or "treat all" situations. Thus, the standardized net benefit indicated lesser risk and more benefit by implementing this model on the same group of population.

## Meta-validation of the ML models on external dataset performances

Meta-validation was performed on two sets of training / external validation datasets, namely, i) 2-four-features (Italian) / 1-four-features (Brazilian) and ii) 3-fourteen-feature (Brazilian) / 1-fourteen-feature (Brazilian). The performances of the ML models were assessed in two dimensions, i) dataset similarity (between training dataset and external validation dataset),

and ii) minimum sample size (MSS); and evaluated in terms of Balanced accuracy, Standardized net benefit and Brier score. The meta-validation results were depicted using External Performance Diagram (Fig 6). The results showed that the minimum sample size exceeded 110% for both the datasets, as apparent from the hue brightness (Fig 6). The 3-fourteen feature/

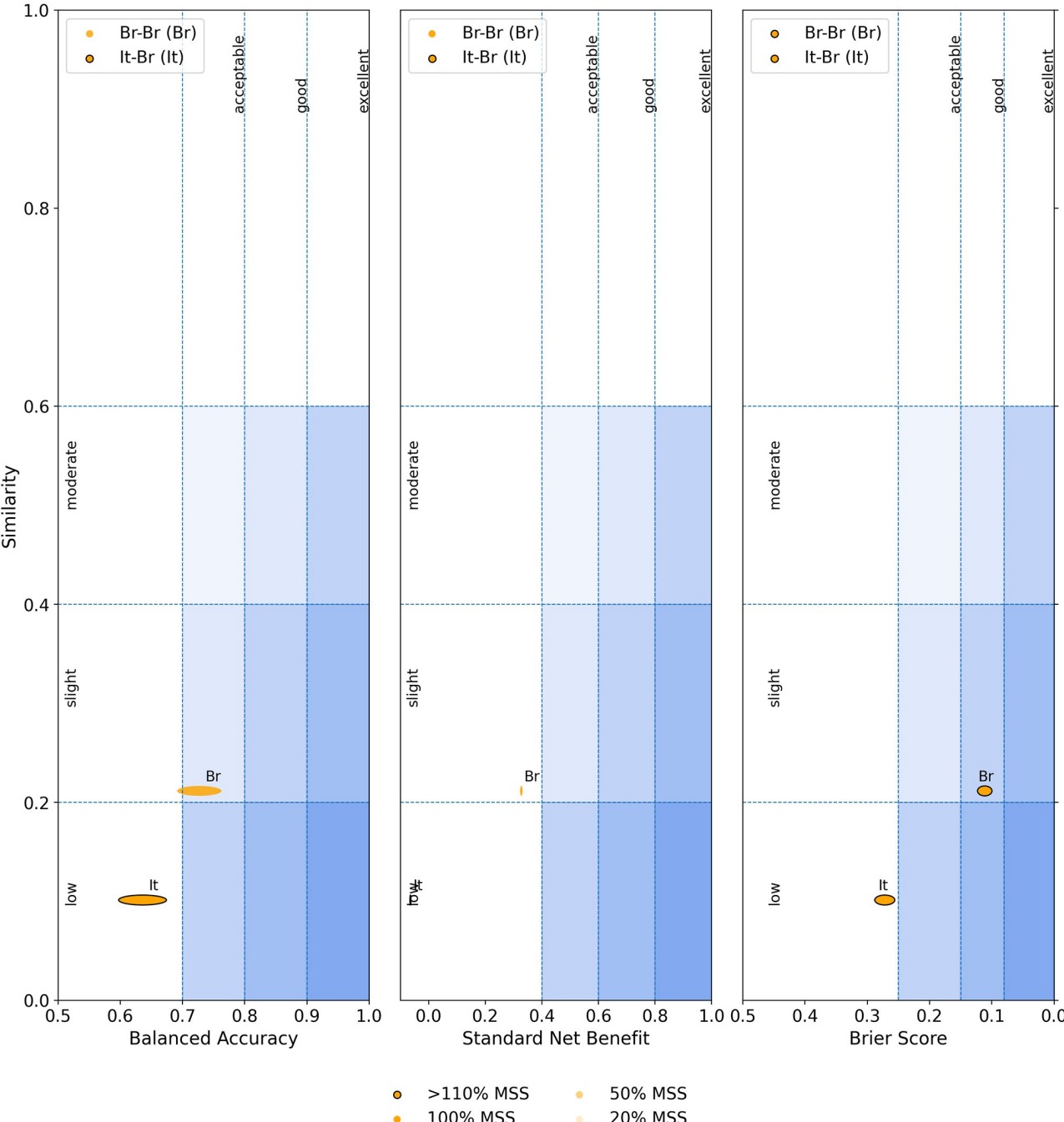

**Fig 6. The external performance diagram generated using the online tool (https://qualiml.pythonanywhere.com), depicted the results from external validation studies on COVID-19 diagnosis trained and tested on i) same population–Brazil-Brazil (Br) and ii) different population–Italy-Brazil (It).** The Minimum sample size, depicted by the hue brightness. The width of the ellipse equals to the width of 95% confidence interval with respect to the given performance metrics.

1-fourteen feature model (training and external validation set both from Brazil) performed well ("good" according to external performance diagram). As reported earlier [36], an external validation can be considered successful when the model exhibits at-least good performance on some of the external datasets. The training and the external validation datasets from Brazil were "slight" in similarity and have adequate sample size (Fig 6). "Slight" similarity (less than 40%) of the external validation dataset with respect to the training dataset implied a reliable test bunch in terms of conservative estimates of model performance [36]. On the other hand, 2-four feature / 1-four feature model (training dataset from Italy and test dataset from Brazil) sample size was adequate according to the hue brightness. The data similarity between training and the test dataset was "low", lower than 20%, hence reliable. However, the model utility was outside the "acceptable" region. According to the previous study [36], there were several discrepancies across Italian and Brazilian datasets those lead to poor performance on their external validation; for example, i) lack of predictive features, such as "Suspect feature" in Brazilian dataset, ii) the instruments used for data collection in these two datasets were different (at least, instrument names were unknown in case of Brazilian datasets), etc.

## Comparison of meta-validation results with the literature reports

In recent past Cabitza et. al. from Italy has performed meta-validation studies for COVID-19 diagnosis based on hematological parameters [36]. Their training dataset comprised of twenty-one hematological features curated from two different hospitals of Italy, along with eight external validation datasets—three from Italy, three from Brazil, one from Spain and one from Ethiopia. The total dataset size (including both training and external validation) was n = 7046, much lower than that presented in this study (n = 195554, raw data). The results from their analyses indicated that the model performance is good or excellent when the external datasets were from Italy. However, the external performances were poorer than acceptable values in case of Spain and Brazil datasets (Fig 8 of ref [36]). The performances were quantified in terms of various metrics, for example, average F2-score for Italian datasets was 87%±3% and that of Brazilian datasets was 37%±4%. This clearly indicated that the model performed good or excellent when trained and tested on the same population (in this case, Italy). Authors have pointed out that the poor performance on external dataset from different population could be due to various reasons, such as, differences in testing equipment, reference ranges, ethnic variability, phenotypic variability, human immune response etc. In the present study, we have considered two external validation datasets, i) training and external dataset from the same population (Brazil) and ii) training (Italy) and external (Brazil) dataset from different populations. The intra-population F2-score was 80% and inter-population F2-score was 49%; the inter-population F2-score, in this study, was better than the published reported (stated above). Similarly, the Brier score reported, in this study (Fig 6), was comparable to the published report (Table 4 of [36]) for intra-population external validation. Whereas, the Brier score for inter-population, reported in this study was lower (0.3) than some of the external performances in the earlier study, indicating better accuracy of the probabilistic prediction. The lower performance of intra-population external validation in this study could be due to the anonymised data in Dataset 1 (obtained from the Kaggle data set) or the missing unit information for different haematological features, in the same dataset. While comparing the utility of the model, in terms of standardized net benefit, the current model performance was poor for the inter-population external validation, at two different probability threshold values (Table 3), compared to the earlier report [36]. One of the possible reasons could be the anonymity of the data and the instrumentation for Dataset 1. The standardized net benefit value for intra-population validation was better at probability threshold value of 0.15, and close to the

**Table 4. XGBoost model performance metrics were shown from one hundred iterations on external validation datasets for a) 2-four-features (Italian) /1-four-features (Brazilian), and b) 3-fourteen-features (Brazilian) / 1-fourteen-feature (Brazilian) with IV-perturbation, and IV-perturbation plus augmentation methods.** Baseline values were reported. The baseline data was generated using lower fuzziness with resampling in a single step, in contrast to perturbation and perturbation plus augmentation data, where higher fuzziness was applied using sequential resampling of the baseline data. The standard deviation values were shown in parentheses.

a)

| Models | AUC | Accuracy | F1 score |
|---|---|---|---|
| IV-perturbation | 0.814 (0.004) | 0.740 (0.005) | 0.750 (0.005) |
| Baseline | 0.695 (0.009) | 0.321 (0.005) | 0.517 (0.007) |
| IV-perturbation plus data augmentation | 0.841 (0.0002) | 0.765 (0.0008) | 0.772 (0.0009) |
| Baseline | 0.731 (0.0008) | 0.327 (0.002) | 0.508 (0.003) |

b)

| Models | AUC | Accuracy | F1 score |
|---|---|---|---|
| IV-perturbation | 0.838 (0.004) | 0.672 (0.007) | 0.428 (0.008) |
| Baseline | 0.841 (0.003) | 0.674 (0.005) | 0.432 (0.006) |
| IV-perturbation plus data augmentation | 0.838 (0.002) | 0.683 (0.004) | 0.422 (0.005) |
| Baseline | 0.839 (0.0003) | 0.680 (0.002) | 0.421 (0.001) |

acceptable range reported earlier [36]. However, the utility of this model was lower compared to the reported models [36]. The limitation could be the unknown instrumentation, units, and normalization schemes used in the Brazilian dataset, that left a gap to compare and calibrate the results. The state-of-the-art (SOTA) of the current study lies in its robustness in the data size (n = 195554) in comparison to the above-mentioned report (n = 7046). Moreover, the current study can predict COVID-19 disease using a lesser number of haematological features (four or fourteen) compared to the previous study (twenty-one features) without compromising the external model performance. In addition, the F2-score and Brier score for the inter-population external validations in the current study were superior to some of those reported earlier. Thus, the current models seem to be more parsimonious and useful toward general applicability.

## Impact of Instantial Variation (IV) on the external validation datasets

Instantial Variation (IV) was defined as the within-subject possible variation that is not due to population differences or errors; rather, intrinsic to a given instance or the measurement process [67]. The biological variations were estimated for different blood parameters, namely leukocytes [68], platelet parameters [69], red blood cell, reticulocyte parameters [70], etc.; where the blood samples were collected from healthy individuals within intervals of weeks (medium-term variations) or intervals of days (short-term variations). The biological variations were marginally different between short-term and medium-term instances, in most of the cases. The impact of IV on the quality of ML models was tested, where the authors showed that ML model performances were poorer for the IV-perturbation data and the model performances improved when IV-perturbation data were augmented with synthetic data [67].

As each instance is different from the other, we studied the effect of IV on the XGBoost model tested on the external validation datasets across a) the Italian and Brazilian populations and b) intra-Brazilian populations. The IV-perturbation and data augmentation were implemented in Python v. 3.10.4, using numpy v. 1.23.0, scikit-learn v. 1.1.1 and scikit-weak v.0.2.0, adapting the zenodo code provided in the literature [67]. The IV-perturbation was performed by resampling each data point following a Gaussian distribution. The Gaussian distribution

width was obtained from the variability scores reported for different hematological parameters [67–70]. One hundred synthetic data points were augmented using the Gaussian distribution for the IV-perturbation plus augmentation method. Three XGBoost performance metrics were reported (Table 4).

The notable point in the performance metrics was very small standard deviation values from one hundred cycles of iterations; more than a logarithmic scale lower than the confidence interval reported elsewhere (Fig 2 of reference 67). Although IV-perturbation has affected the model performance on the inter-population data (Italy versus Brazil), the intra-population data was least affected (Table 4). Thus, the XGBoost model was robust enough on the external validation dataset from the same population which indicated its potential applicability. There could be three possible reasons for less IV-perturbation of the XGBoost model (at least for one dataset) compared to the literature report [67], i) the external validation data size was sufficiently large, ii) the data were internally normalized (as mentioned in the method section), and inherent robustness of the XGBoost model.

## Blind prediction of XGBoost models with four hematological parameters on West European populations

To further validate the efficacy of the working models, we have considered one more dataset from published literature with thirty-seven features, including the data points from different stages (time points) of COVID-19 [25]. The dataset was obtained from the literature without pre-processing (no feature, record, or data point removed). The source authors have reported two distinct stages of COVID-19 patients, W.E.-*early* and W.E.-*advanced*. Distributions of four haematological parameters across the datasets, 1-four-features, 2-four-features, W.E.-*early*, and W.E.-*advanced*, were compared (Fig 7). The distributions were almost the same across all the datasets for Leukocytes and platelets. For eosinophils and monocytes, the distributions for datasets 2-four-features and W.E.-*early* are similar, and for the distributions across datasets 1-four-features and W.E.-*advanced*. The external performance of the model on the W.E.-*early* dataset (0.65) was high compared to that on the W.E.-*advanced* dataset (0.52) (Table 5). To note, W.E.-*early* and W.E.-*advanced* datasets contain information only from COVID-19 patients and no negative controls. Hence, only the sensitivity metric was reported (Table 5).

## Deployment of prediction server

We deployed a web server where two sets of inputs are accepted for binary COVID-19 prediction, i) four hematological parameters (leukocyte, monocyte, eosinophil, and platelet count) and ii) fourteen-parameter models (CBC and WBC differentials). The server outputs the COVID-9 results, either positive or negative, with the COVID-19 probability reported in percentage. Link to the web server: https://covipred.bits-hyderabad.ac.in/home. Design of the webserver is given the supplementary information.

## Conclusion

Considering the need to develop an alternate protocol for rapid, near-accurate, and cheaper COVID-19 detection technique, we aimed to externally validate the haematology-based ML prediction by optimizing the features, which is yet to be fully understood. We have integrated published clinical records from Brazil, Italy, and West Europe hospitals. The data from Brazil and Italy were classified into four sub-datasets and trained on seven different ML methods. The XGBoost algorithm consistently performed superior to other ML methods. The internal performances of the XGBoost models were compared with the published reports available on the same datasets; the models reported here outperformed the published reports. The meta

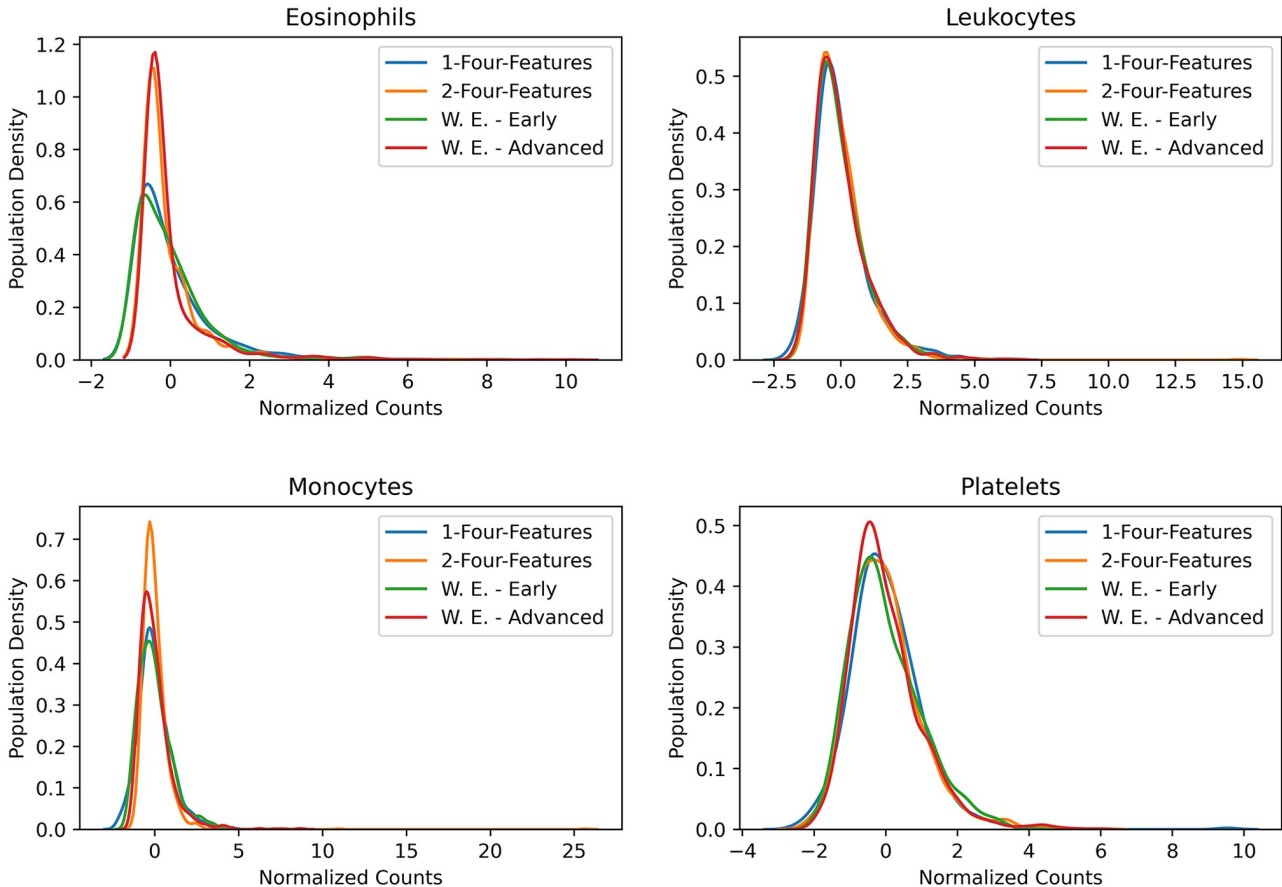

**Fig 7. Distributions of four hematological parameters across four different datasets (two training datasets–Dataset 1-four-features and Dataset 2-four-features and two test datasets–*early* and *advance*).** The hematological parameters are–a) platelet, b) leukocyte c) eosinophil and d) monocyte. These distributions indicate the proximity of the individual test datasets to the training datasets.

validation of the ML models on external datasets indicated the acceptability of the external performances; these results were either comparable or superior to the published reports. In this study, two sets of haematological parameters were selected for ML models, i) four features–leukocytes, monocytes, platelets and eosinophils (those were common across three different countries, namely, Brazil, Italy and India) and ii) fourteen features–CBC and WBC differentials. Both the set of parameters are represented by basic routine blood tests, usually, available in low-resource settings (CBC dataset). The ML methodology developed and externally validated here, was based on routine blood examination outcomes–available for inpatients and emergency-admitted patients. The current models, developed on a large sample cohort, can be used parsimoniously (as it used lesser number of haematological features compared to the

**Table 5. Blind prediction of XGBoost model trained on dataset 2-four-feature and tested on W.E.-*early* and W. E.-*advanced* datasets.** The *early* and *advanced* datasets contain only COVID-19-positive patient results; no negatives were available. Hence, only sensitivity values reported.

| Training set/test set | Sensitivity |
|---|---|
| Dataset 2-four-feature/ W.E.-*early* | 0.65 |
| Dataset 2-four-feature/W.E.-*advanced* | 0.52 |

previous study) for COVID-19 diagnosis. Moreover, the XGBoost model was marginally perturbed by Instantial Variations (IV), albeit noise, in the intra-population external validation dataset. To note, ML models have their own limitations, in terms of dataset dependencies, size of the datasets, ethnic variabilities, phenotypic variabilities, analytical instrumentations for clinical chemistry tests, etc. This shortcoming was, to some extent, reflected in the external validation; for four-feature models—specificity value was good but sensitivity value was poor (Table 3a). As reviewed earlier, variation in analytical instrumentations may exhibit extreme heterogeneity, thus may cause concern for the further usage of the model [66]. On the other hand, external validation for fourteen-feature model exhibited good sensitivity but poor specificity values (Table 3b). One way to overcome this limitation would be using the blood test results from same analytical instrumentations. However, as suggested in the literature, CBC standardization was less problematic due to change in analytical instrumentations compared to other tests [23]. Nevertheless, ML-based models are low cost and depends on rapid blood test exam, providing a good start for initial screening. Moreover, these results could be combined and compounded with qRT-PCR tests with an expectation of higher accuracy and sensitivity for the suspected cases. Thus, large scale identification of COVID-19 patients can be done in timely manner. Two XGBoost models, based on these two sets of features, were selected for external evaluations. The external performance of the fourteen-parameter XGBoost model trained and tested on the Brazilian dataset was comparable to that of the internal performance. However, the external performances of the four-parameter XGBoost model trained on the Italian dataset and tested on a) Brazilian and b) West European datasets were poorer than the internal evaluation. The results promise the utility of these models when trained and tested on the same populations. However, it also warns to use the model, with caution, trained on one population and tested on another. The outcome of this work has the potential for an initial screen of COVID-19 based on haematological parameters. In future work, we aim to train and test those on the Indian population to use at the healthcare centres of India.

## Supporting information

**S1 Appendix. 5 supporting tables and additional description on the webserver development.**
(DOCX)

## Author Contributions

**Conceptualization:** Sibnath Ray, Debashree Bandyopadhyay.

**Data curation:** Ali Safdari, Chanda Sai Keshav, Kshitij Verma, Vaadeendra Kumar Burra.

**Formal analysis:** Ali Safdari, Deepanshu Mody, Kshitij Verma, Utsav Kaushal, Vaadeendra Kumar Burra, Sibnath Ray.

**Funding acquisition:** Debashree Bandyopadhyay.

**Investigation:** Sibnath Ray, Debashree Bandyopadhyay.

**Methodology:** Chanda Sai Keshav, Deepanshu Mody, Utsav Kaushal, Vaadeendra Kumar Burra.

**Project administration:** Debashree Bandyopadhyay.

**Resources:** Debashree Bandyopadhyay.

**Software:** Ali Safdari, Chanda Sai Keshav, Deepanshu Mody, Utsav Kaushal, Vaadeendra Kumar Burra, Debashree Bandyopadhyay.

**Supervision:** Debashree Bandyopadhyay.

**Validation:** Deepanshu Mody, Utsav Kaushal, Vaadeendra Kumar Burra, Debashree Bandyopadhyay.

**Visualization:** Utsav Kaushal, Vaadeendra Kumar Burra.

**Writing – original draft:** Ali Safdari, Vaadeendra Kumar Burra.

**Writing – review & editing:** Debashree Bandyopadhyay.

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
