## [Decision Letter · Decision Letter 0]

28 Jul 2023

PONE-D-23-15648The external validity of machine learning-based prediction scores from hematological parameters of COVID-19: A study using hospital records from Brazil, Italy, and Western EuropePLOS ONE

Dear Dr. Bandyopadhyay,

Thank you for submitting your manuscript to PLOS ONE. After careful consideration, we feel that it has merit but does not fully meet PLOS ONE’s publication criteria as it currently stands. Therefore, we invite you to submit a revised version of the manuscript that addresses the points raised during the review process.

We look forward to receiving your revised manuscript.

Kind regards,

Francesco Branda

Academic Editor

PLOS ONE

Journal Requirements:

Reviewers' comments:

Reviewer's Responses to Questions

**Comments to the Author**

1. Is the manuscript technically sound, and do the data support the conclusions?

Reviewer #1: Partly

Reviewer #2: Yes

Reviewer #3: Yes

Reviewer #4: No

2. Has the statistical analysis been performed appropriately and rigorously? 

Reviewer #1: No

Reviewer #2: Yes

Reviewer #3: Yes

Reviewer #4: No

3. Have the authors made all data underlying the findings in their manuscript fully available?

Reviewer #1: Yes

Reviewer #2: Yes

Reviewer #3: Yes

Reviewer #4: No

4. Is the manuscript presented in an intelligible fashion and written in standard English?

Reviewer #1: Yes

Reviewer #2: Yes

Reviewer #3: Yes

Reviewer #4: Yes

5. Review Comments to the Author

Reviewer #1: The following points should be considered in the manuscript:

1-The authors have focused on specific type of the haematological parameters without adding any explanations about the reasons of the using. So, the authors must explain this significant point in details.

2-The authors must explain in more and more details how they got the actual positive results of applying the model which they focused on in the manuscript on the biological tissues effects.

3-The authors must add more details and information about the limitations of the methodology which they have reached in the manuscript where there are multiple studies represent how the same methodology appears its limitations because this is a significant point.

4-In the figure 6, the authors must retype the actual units of the measurement.

5-The figure 2, 3 and 4 in addition to the related results of the manuscript need more comparisons in the light of the methodology itself.

6-The availability of the methodology which the authors have used in the manuscript must be expanded to other problems.

7-In the conclusion section of the manuscript, the authors must add more details about the future improvements of the suggested methodology in the light of the methodology of the manuscript and about its availability to other measuring problems.

8-There are some typographical or grammatical mistakes found in the text of the manuscript, so, the authors must recheck all of those mistakes over all parts of the manuscript.

9-There are some recent literature studies about the transmission of COVID-19 in Brazil using of the machine learning techniques which is doi:10.1515/em-2021-0029, and doi:10.1515/em-2020-0036, and doi:10.1515/em-2022-0108, and the authors must discuss those literature studies as examples of those studies.

10-The statistical analyses of the methodology for the rat is pure, it is preferable mention the positive and the negative point for each case.

11-There are some indicators of the used methodology are not included where there are multiple effects of those indicators.

12-In the table 1 of the manuscript, there is missing discussions of some significant details, and it must be taken into account.

13-the table 2 of the manuscript needs more discussion because this table is a key-role of the results of the manuscript.

14-The availability of the methodology procedures of the manuscript which the authors have employed must be expanded by adding more information and details.

15-There are some symbols or abbreviations used in the text of the manuscript without any definition, so the authors must define each symbol or abbreviation even if the symbol is very well known.

16-The authors must show the negative effects of the ML model.

17-The authors must retype all of the equations of the manuscript in a more effective way.

Reviewer #2: Introduction

Early detection of severely infected COVID-19 patients, ICU admission, and comprehensive information from previous studies on mortality estimation are important for understanding the importance of the subject. This information to be added will increase the validity and reliability of the study. In addition, this study draws attention to some biomarkers in the diagnosis and determination of the severity of COVID-19. Your reason for choosing these features will be better understood if you consider the explanations I have suggested below. Pay attention to these suggestions.

First of all, in the introduction of this study, a paragraph should be written about the importance of routine blood values, which are effective in the diagnosis, prognosis and mortality of COVID-19.

1) There is important information in the following articles that point to the role and importance of laboratory markers in the diagnosis, prognosis and mortality of COVID-19. I suggest citing the key findings of these articles.

-What Is the Impact and Efficacy of Routine Immunological, Biochemical and Hematological Biomarkers as Predictors of COVID-19 Mortality? Int. Immunopharmacol. 2022, 105, 108542. https://doi.org/10.1016/j.intimp.2022.108542.

-The Effectiveness of Blood Routine Parameters and Some Biomarkers as a Potential Diagnostic Tool in the Diagnosis and Prognosis of Covid-19 Disease. Int. Immunopharmacol. 2021, 98, 107838. https://doi.org/10.1016/j.intimp.2021.107838.

-Forecasting of Oxidant/Antioxidant levels of COVID-19 patients by using Expert models with biomarkers used in the Diagnosis/Prognosis of COVID-19. Int. Immunopharmacol., 2021; 100, 108127. https://doi.org/10.1016/j.intimp.2021.108127.

2) There is important information in the following articles emphasizing the changes in routine blood values and some demographic characteristics in the deterioration of immune responses and the increase in the severity and mortality of the disease in COVID-19. I suggest citing the key findings of these articles.

-COVID-19 Is More Dangerous for Older People and Its Severity Is Increasing: A Case-Control Study. Med. Gas Res. 2022, 12, 51–54. https://doi.org/ 10.4103/2045-9912.325992.

-How Do Routine Laboratory Tests Change in Coronavirus Disease 2019? Scand. J. Clin. Lab. Investig. 2021, 81, 24–33. https://doi.org/10.1080/00365513.2020.1855470.

-Oxyhemoglobin Dissociation Curve in COVID-19 Patients. Meandros Med Dent J 2023;24(1):58-64. https://doi.org/10.4274/meandros.galenos.2023.87049.

-Prediction of Diagnosis and Prognosis of COVID-19 Disease by Blood Gas Parameters Using Decision Trees Machine Learning Model: A Retrospective Observational Study. Med. Gas Res. 2022, 12, 60–66. https://doi.org/10.4103/2045-9912.326002.

3) The following articles describe the efficacy of important biomarkers as a potential predictor of diagnosis, prognosis and mortality of COVID-19. I propose to refer to the main findings of these articles.

-Diagnosis and Prognosis of COVID-19 Disease Using Routine Blood Values and LogNNet Neural Network. Sensors 2022, 22, 4820. https://doi.org/10.3390/s22134820.

-Automatic Detection of Severely and Mildly Infected COVID-19 Patients with Supervised Machine Learning Models. IRBM 2022, 1, 1–12. https://doi.org/10.1016/j.irbm.2022.05.006.

-Detection of Risk Predictors of COVID-19 Mortality with Classifier Machine Learning Models Operated with Routine Laboratory Biomarkers. Appl. Sci. 2022, 12, 12180. https://doi.org/10.3390/app122312180.

-Machine Learning Sensors for Diagnosis of COVID-19 Disease Using Routine Blood Values for Internet of Things Application. Sensors 2022, 22, 7886. https://doi.org/10.3390/s22207886.

-Effect of ferritin, INR, and D-dimer immunological parameters levels as predictors of COVID-19 mortality: A strong prediction with the decision trees. Heliyon, e14015. https://doi.org/10.1016/j.heliyon.2023.e14015.

4) Findings should not be mentioned in the introduction. The sentences that are suitable for the hypothesis and the purpose of the article are sufficient.

Material and Method

-Information about patient exclusion criteria and demographic characteristics should be given.

-Which features are taken should be stated in units of these features. This is important for clinicians.

-I suggest that the explanations of machine learning algorithms should be given as sub-headings. In addition, literature should be given for these explanations.

-Is the feature selection model used for machine learning models? should be specified.

- It should be stated what the performance metrics are used for (for example, for diagnostic performance).

Results

- 95% confidence limits of performance metrics should be given.

-There is a big difference between sensitivity and specificity results in model performance results. This indicates the imbalance of the data set. Explain what was done to balance and optimize the dataset.

- It will be more beautiful if the figures are colored (especially figure 4). In addition, the resolution of the figure should be increased.

Dıscussıon and Conclusion

- I could not see a comparison of the important findings in this study with the literature. Discussion on this should be added. You can also use the above-mentioned articles for this.

-Which features were most effective in the diagnosis of the disease? these should be specified.

- The prominent findings of the article should be reported in articles.

Reviewer #3: The external validity of machine learning-based prediction scores from hematological parameters of COVID-19: A study using hospital records from Brazil, Italy, and Western Europe

The manuscript proposes an interesting investigation on the performance of machine learning techniques to predict scores from hematological parameters of COVID-19. The study also compares prediction performance on data from different countries. The paper is well structured and discussed. Results are properly presented and explained. Here are some questions/recommendations to improve its quality.

• In line 219, authors state that an imputation process was required for ML models other than XGBoost. Was any statistical test conducted to ensure that the imputed information did not change the underlying structure of the data?

• What is the proportion between training and testing sets? How was that splitting performed? Could different proportion between training and testing affect the performance of the classifier?

• In line 254 it is mentioned that hematological features were selected based on the Pearson correlation coefficients between the features and the SARS-COV-2 results (positive or negative). Any concern on the correlation among the features themselves? Several classifiers tend to have their performance reduced when dealing with highly correlated features. Please comment on it. Also, were these retained features meaningful from a practical perspective?

• How were the cutting limits for the Pearson correlation defined?

• I would recommend a minor comment on the computational processing time required by each technique (if possible), since more complex techniques tend to be very time-consuming.

• Results on Table 3 a) are substantially smaller than previous ones. Please provide further explanation on such lower performance metrics.

Reviewer #4: The paper presents an application of Machine Learning methodologies to predict positivity to SARS-CoV-2 based on blood exams results.

I appreciated the attempt to include a diverse set of countries and contexts. Nonetheless, I believe this work has significant limitations.

Major comments

1) I find it puzzling why correlations are used a priori to select features. Models like XGBoost work well also with a large number of features and are capable of automatically selecting the most relevant ones or identifying meaningful combinations. Relying solely on linear pair-wise correlation coefficients to exclude features based on their relationship with the outcome might hinder the discovery of valuable interaction patterns. These interactions may involve features that lack significant predictive power individually but demonstrate strong predictive ability when considered in combination with other features

2) The Pearson correlation coefficient is not the most appropriate measure to compute the correlation between a binary variable and a continuous or categorical variable. Indeed, the Pearson correlation coefficient is specifically designed to measure the linear relationship between two continuous variables. More suitable approaches should be considered instead (point-biserial correlation or biserial correlation, other non parametric tests).

3) The motivation of the paper appears to be insufficiently convincing. While the paper emphasises the advantages of haematological analysis, it overlooks significant limitations. For instance, the paper fails to acknowledge that RT-PCR, despite being more expensive, allows for simpler sample collection through nasal or oral swabs, unlike haematological analysis, which requires a blood sample. Additionally, the paper neglects to address rapid tests, a relevant aspect that warrants discussion. Including a comprehensive analysis of different testing methods and their respective pros and cons would significantly enhance the paper's credibility and provide a more balanced perspective.

4) Regarding the last point, the current framing appears to have a significant issue. The majority of the findings in the presentation revolve around the model's performance, which raises concerns. While the model's performance may be highlighted, it fails to demonstrate the practical advantages of using such a model. There seem to be limited benefits to adopting this model when individuals could obtain similar, if not more accurate, results using a rapid test or PCR without the need for blood collection. Additionally, the paper indicates that the model's performance dramatically declines when trained on a different population. The information presented in table 3A and 3B suggests that the model behaves similarly to a random classifier in such cases. This is a critical limitation as it contradicts the emphasis on the model's performance. It becomes challenging to justify the practical application of this model when its effectiveness diminishes significantly when applied to different populations. See for example the last sentence of the abstract “Thus, current models can be applied at other demographic locations.” I find it very hard to support this statement looking at the results from Table 3.

Minor comments

1) The description of the Web application is not very useful and do not add much to the paper, therefore I believe it should be shortened significantly and moved eventually in the Supplementary Information

2) The dataset description is quite hard to follow, why so many variations of the same dataset are produced? I’d suggest presenting the results for selected datasets in the main, and move the rest to the supplementary information. Also, there are many arbitrary decisions in the dataset creation part that need justifications (examples are the threshold of null values used to drop entries which are not even consistent across dataset, in some cases it is 90%, in others 66%, why?)

3) References to ML models and packages used should be included

4) I could not find details on model training criteria (cross-validation, parameter selection, etc)

6. PLOS authors have the option to publish the peer review history of their article (what does this mean?). If published, this will include your full peer review and any attached files.

Reviewer #1: No

Reviewer #2: No

Reviewer #3: No

Reviewer #4: No

---

## [Author Response · Author response to Decision Letter 0]

18 Nov 2023

Reviewer #1: The following points should be considered in the manuscript:

1-The authors have focused on specific type of the haematological parameters without adding any explanations about the reasons of the using. So, the authors must explain this significant point in details.

Response: The explanation for choosing specific hematological parameters were added in “Results and Discussion” section, subheading – “Correlation between features and SARS-COV-2 results in different clinical datasets”, line number (version with all-markup):370-442 and 453-468.

2-The authors must explain in more and more details how they got the actual positive results of applying the model which they focused on in the manuscript on the biological tissues effects.

Response: To the best of our knowledge, biological tissue effects were not studied in this work. 

Here we suppose “actual positive results” indicate the “ground truth” used in this study for comparison with the predicted results. The “actual positive results” were obtained from molecular tests, RT-PCR results, from different hospitals. More explanations are added in line number (version with all markups) 150-156; 202-209; 232-233.

3-The authors must add more details and information about the limitations of the methodology which they have reached in the manuscript where there are multiple studies represent how the same methodology appears its limitations because this is a significant point.

Response: More detailed information added in line number (version with all markups) 569-578

4-In the figure 6, the authors must retype the actual units of the measurement.

Response: Actual units of measurements were added in Figure 2. In Figure 6, normalized counts were shown.

5-The figure 2, 3 and 4 in addition to the related results of the manuscript need more comparisons in the light of the methodology itself.

Response: More comparison added for -

Figure 2, line number (version with all markups) 372-385; 395-400; 410-417; 424-439.

Figure 3, line number (version with all markups) 445-468

Figure 4, line number (version with all markups) 487-497

6-The availability of the methodology which the authors have used in the manuscript must be expanded to other problems.

Response: The XGBoost models selected from internal evaluation were expanded to external evaluation, line number (version with all markups) 586-590; and blind predictions line number (601-610)

7-In the conclusion section of the manuscript, the authors must add more details about the future improvements of the suggested methodology in the light of the methodology of the manuscript and about its availability to other measuring problems.

Response: Suggestions incorporated. Line number (Version with all markups) 664-667

8-There are some typographical or grammatical mistakes found in the text of the manuscript, so, the authors must recheck all of those mistakes over all parts of the manuscript.

Response: The typographical or grammatical mistakes were rechecked manually and with Grammarly (https://app.grammarly.com/apps).

9-There are some recent literature studies about the transmission of COVID-19 in Brazil using of the machine learning techniques which is doi:10.1515/em-2021-0029, and doi:10.1515/em-2020-0036, and doi:10.1515/em-2022-0108, and the authors must discuss those literature studies as examples of those studies.

Response: References included, line number (version with all markups) 104-106, reference numbers; - 27 and 28. The third reference suggested was not aligned to the current work – not included.

10-The statistical analyses of the methodology for the rat is pure, it is preferable mention the positive and the negative point for each case.

Response: Suggestions included, line number (version with all markups) 58-79

11-There are some indicators of the used methodology are not included where there are multiple effects of those indicators.

Response: Authors assumed that the reviewer mentioned indicators as hematological parameters. Those discussions were incorporated in line number (version with all markups) 91-100; 117-121.

12-In the table 1 of the manuscript, there is missing discussions of some significant details, and it must be taken into account.

Response: Discussions added, line number (version with all markups) 166-170; 211-216

13-the table 2 of the manuscript needs more discussion because this table is a key-role of the results of the manuscript.

Response: Discussion added, line number (version with all markups) 528-555

14-The availability of the methodology procedures of the manuscript which the authors have employed must be expanded by adding more information and details.

Response: More information and references were added, reference numbers 42-48. 

15-There are some symbols or abbreviations used in the text of the manuscript without any definition, so the authors must define each symbol or abbreviation even if the symbol is very well known.

Response: Abbreviations added at the end, line number (version with all markups) 682:701

16-The authors must show the negative effects of the ML model.

Response: Suggestions incorporated. Line number (version with all markups) 659-664

17-The authors must retype all of the equations of the manuscript in a more effective way.

Response: Suggestions incorporated. Line number (version with all markups) 349-351, Eq 1-3.

Reviewer #2: Introduction

Early detection of severely infected COVID-19 patients, ICU admission, and comprehensive information from previous studies on mortality estimation are important for understanding the importance of the subject. This information to be added will increase the validity and reliability of the study. In addition, this study draws attention to some biomarkers in the diagnosis and determination of the severity of COVID-19. Your reason for choosing these features will be better understood if you consider the explanations I have suggested below. Pay attention to these suggestions.

First of all, in the introduction of this study, a paragraph should be written about the importance of routine blood values, which are effective in the diagnosis, prognosis and mortality of COVID-19.

Response: Suggestions incorporated line number (version with all markups) 83-100

1) There is important information in the following articles that point to the role and importance of laboratory markers in the diagnosis, prognosis and mortality of COVID-19. I suggest citing the key findings of these articles.

-What Is the Impact and Efficacy of Routine Immunological, Biochemical and Hematological Biomarkers as Predictors of COVID-19 Mortality? Int. Immunopharmacol. 2022, 105, 108542. https://doi.org/10.1016/j.intimp.2022.108542.

Response: Reference incorporated, reference no., 13

-The Effectiveness of Blood Routine Parameters and Some Biomarkers as a Potential Diagnostic Tool in the Diagnosis and Prognosis of Covid-19 Disease. Int. Immunopharmacol. 2021, 98, 107838. https://doi.org/10.1016/j.intimp.2021.107838.

Response: Reference incorporated, reference no., 19

-Forecasting of Oxidant/Antioxidant levels of COVID-19 patients by using Expert models with biomarkers used in the Diagnosis/Prognosis of COVID-19. Int. Immunopharmacol., 2021; 100, 108127. https://doi.org/10.1016/j.intimp.2021.108127.

Response: Reference incorporated, reference no., 20

2) There is important information in the following articles emphasizing the changes in routine blood values and some demographic characteristics in the deterioration of immune responses and the increase in the severity and mortality of the disease in COVID-19. I suggest citing the key findings of these articles.

-COVID-19 Is More Dangerous for Older People and Its Severity Is Increasing: A Case-Control Study. Med. Gas Res. 2022, 12, 51–54. https://doi.org/ 10.4103/2045-9912.325992.

Response: Reference incorporated, reference no., 17

-How Do Routine Laboratory Tests Change in Coronavirus Disease 2019? Scand. J. Clin. Lab. Investig. 2021, 81, 24–33. https://doi.org/10.1080/00365513.2020.1855470.

Response: Reference incorporated, reference no., 40

-Oxyhemoglobin Dissociation Curve in COVID-19 Patients. Meandros Med Dent J 2023;24(1):58-64. https://doi.org/10.4274/meandros.galenos.2023.87049.

Response: Reference incorporated, reference no. 41

-Prediction of Diagnosis and Prognosis of COVID-19 Disease by Blood Gas Parameters Using Decision Trees Machine Learning Model: A Retrospective Observational Study. Med. Gas Res. 2022, 12, 60–66. https://doi.org/10.4103/2045-9912.326002.

Response: Reference incorporated, reference no., 29

3) The following articles describe the efficacy of important biomarkers as a potential predictor of diagnosis, prognosis and mortality of COVID-19. I propose to refer to the main findings of these articles.

-Diagnosis and Prognosis of COVID-19 Disease Using Routine Blood Values and LogNNet Neural Network. Sensors 2022, 22, 4820. https://doi.org/10.3390/s22134820.

Response: Reference incorporated, reference no., 30

-Automatic Detection of Severely and Mildly Infected COVID-19 Patients with Supervised Machine Learning Models. IRBM 2022, 1, 1–12. https://doi.org/10.1016/j.irbm.2022.05.006.

Response: Reference incorporated, reference no., 31

-Detection of Risk Predictors of COVID-19 Mortality with Classifier Machine Learning Models Operated with Routine Laboratory Biomarkers. Appl. Sci. 2022, 12, 12180. https://doi.org/10.3390/app122312180.

Response: Reference incorporated, reference no., 32

-Machine Learning Sensors for Diagnosis of COVID-19 Disease Using Routine Blood Values for Internet of Things Application. Sensors 2022, 22, 7886. https://doi.org/10.3390/s22207886.

Response: Reference incorporated, reference no., 33 

-Effect of ferritin, INR, and D-dimer immunological parameters levels as predictors of COVID-19 mortality: A strong prediction with the decision trees. Heliyon, e14015. https://doi.org/10.1016/j.heliyon.2023.e14015.

Response: Reference incorporated, reference no., 34

4) Findings should not be mentioned in the introduction. The sentences that are suitable for the hypothesis and the purpose of the article are sufficient.

Response: Suggestions incorporated, line number (version with all markups) 107-111

Material and Method

-Information about patient exclusion criteria and demographic characteristics should be given.

Response: Suggestions incorporated, line number (version with all markups) 152-154; 202-209; 241-243

-Which features are taken should be stated in units of these features. This is important for clinicians.

Response: Unit were added in Figure 2

-I suggest that the explanations of machine learning algorithms should be given as sub-headings. In addition, literature should be given for these explanations.

Response: Sub-headings added for individual machine learning algorithms in method section

-Is the feature selection model used for machine learning models? should be specified.

Response: Feature selection was revisited. In the revised version feature selection was based on the common parameters across different countries, line number: 431-442

- It should be stated what the performance metrics are used for (for example, for diagnostic performance).

Response: Suggestion incorporated, line number 341-345; 360-361

Results

- 95% confidence limits of performance metrics should be given.

Response: Suggestion incorporated, line number 350-353 (Eq. 4), Table 2 and Table 3.

-There is a big difference between sensitivity and specificity results in model performance results. This indicates the imbalance of the data set. Explain what was done to balance and optimize the dataset.

Response: XGBoost, being a ternary classifier, can handle data imbalance on its own. For other models, data imputation was done. 

There is a possibility to improve poor sensitivity and specificity by using the training and test dataset from the same analytical instrumentation (line number:664-665). However, implementation of the same is beyond the scope of the study.

- It will be more beautiful if the figures are colored (especially figure 4). In addition, the resolution of the figure should be increased.

Response: All the figures are now in colour

Dıscussıon and Conclusion

- I could not see a comparison of the important findings in this study with the literature. Discussion on this should be added. You can also use the above-mentioned articles for this.

Response: Suggestions incorporated – “Results and Discussions” – subsection “Comparison of internal performances of the XGBoost model with published reports”, line number, 525-555

-Which features were most effective in the diagnosis of the disease? these should be specified.

Response: Suggestion incorporated – “Conclusions” – line number: 652-657

- The prominent findings of the article should be reported in articles.

Response: Suggestion incorporated – “Conclusions” – line number: 652-671

Reviewer #3: The external validity of machine learning-based prediction scores from hematological parameters of COVID-19: A study using hospital records from Brazil, Italy, and Western Europe

The manuscript proposes an interesting investigation on the performance of machine learning techniques to predict scores from hematological parameters of COVID-19. The study also compares prediction performance on data from different countries. The paper is well structured and discussed. Results are properly presented and explained. Here are some questions/recommendations to improve its quality.

• In line 219, authors state that an imputation process was required for ML models other than XGBoost. Was any statistical test conducted to ensure that the imputed information did not change the underlying structure of the data?

Response: Chi-squared test was performed – “Method” section, line number 330-336. 

• What is the proportion between training and testing sets? How was that splitting performed? Could different proportion between training and testing affect the performance of the classifier?

Response: Proportion of training and testing sets was 90:10 (line no. 308-309). A 10-fold cross-validation was performed, which showed little effect on the classifier's performance with different sets of training and testing data, line number 485-487 and Table S4.

• In line 254 it is mentioned that hematological features were selected based on the Pearson correlation coefficients between the features and the SARS-COV-2 results (positive or negative). Any concern on the correlation among the features themselves? Several classifiers tend to have their performance reduced when dealing with highly correlated features. Please comment on it. Also, were these retained features meaningful from a practical perspective?

Response: Pearson correlation coefficient was replaced by point biserial coefficients. In the revised version of the manuscript, the hematological features were not selected based on correlation coefficients only. Selection of the features were described in details in Results and Discussion” section – sub-heading – “Correlation between features and SARS-COV-2 results in different clinical datasets” – line number: 370-442

• How were the cutting limits for the Pearson correlation defined?

Response: Not relevant in this revised version.

• I would recommend a minor comment on the computational processing time required by each technique (if possible), since more complex techniques tend to be very time-consuming.

Response: Elapsed time for different models were reported in Table S6; line number: 500-502

• Results on Table 3 a) are substantially smaller than previous ones. Please provide further explanation on such lower performance metrics.

Response: Suggestion incorporated in “Conclusion” – line number 659-664

Reviewer #4: The paper presents an application of Machine Learning methodologies to predict positivity to SARS-CoV-2 based on blood exams results.

I appreciated the attempt to include a diverse set of countries and contexts. Nonetheless, I believe this work has significant limitations.

Major comments

1) I find it puzzling why correlations are used a priori to select features. Models like XGBoost work well also with a large number of features and are capable of automatically selecting the most relevant ones or identifying meaningful combinati

---

## [Decision Letter · Decision Letter 1]

9 Jan 2024

PONE-D-23-15648R1The external validity of machine learning-based prediction scores from hematological parameters of COVID-19: A study using hospital records from Brazil, Italy, and Western EuropePLOS ONE

Dear Dr. Bandyopadhyay,

Thank you for submitting your manuscript to PLOS ONE. After careful consideration, we feel that it has merit but does not fully meet PLOS ONE’s publication criteria as it currently stands. Therefore, we invite you to submit a revised version of the manuscript that addresses the points raised during the review process.

We look forward to receiving your revised manuscript.

Kind regards,

Francesco Branda, Ph.D.

Academic Editor

PLOS ONE

Comments from PLOS Editorial Office: We note that one or more reviewers has recommended that you cite specific previously published works. As always, we recommend that you please review and evaluate the requested works to determine whether they are relevant and should be cited. It is not a requirement to cite these works. We appreciate your attention to this request.

**Additional Editor Comments:**

In summary, the major revisions suggested include addressing the methodological and statistical aspects, improving the introduction and discussion by incorporating relevant literature, clarifying dataset descriptions, and addressing practical limitations and motivations for the study.

Reviewers' comments:

Reviewer's Responses to Questions

**Comments to the Author**

1. If the authors have adequately addressed your comments raised in a previous round of review and you feel that this manuscript is now acceptable for publication, you may indicate that here to bypass the “Comments to the Author” section, enter your conflict of interest statement in the “Confidential to Editor” section, and submit your "Accept" recommendation.

Reviewer #5: (No Response)

Reviewer #6: All comments have been addressed

2. Is the manuscript technically sound, and do the data support the conclusions?

Reviewer #5: (No Response)

Reviewer #6: Yes

3. Has the statistical analysis been performed appropriately and rigorously? 

Reviewer #5: (No Response)

Reviewer #6: Yes

4. Have the authors made all data underlying the findings in their manuscript fully available?

Reviewer #5: (No Response)

Reviewer #6: No

5. Is the manuscript presented in an intelligible fashion and written in standard English?

Reviewer #5: (No Response)

Reviewer #6: Yes

6. Review Comments to the Author

Reviewer #5: The research is significant as it addresses the need for rapid, accurate, and cost-effective COVID-19 detection methods, especially in large-scale and low-income populations.

However, the manuscript, even in its revised version, still has numerous flaws that need to be addressed to allow for its publication.

1) The authors claim that used an innovative approach combining machine learning with hematological parameters for COVID-19 prediction, but they do not consider previous studies that have already been published on this matter as published in [1]. Again, authors claim that the novelty lies in the external validation of the ML models across different populations and countries, which has not been explored in previous studies. Unfortunately, they do not consider [2], a study published two years ago with different dataset collected across 3 different continents.

2) The diminished effectiveness of the model when trained with data from one population and applied to another raises concerns about its wider applicability. The authors mention that “the combination of CBC parameters varies with ethnicity” (line 98), yet they provide no further elaboration or references to support this assertion. It's crucial, in my view, for the authors to delve deeper into this premise, as it underpins the main argument against the reliability of tools utilizing laboratory data across diverse datasets. I highly recommend a thorough review of reference [3].

3) Another significant limitation of this paper is the complete absence of details about the methodologies employed for the laboratory tests. While it's acknowledged that hematological instruments tend to be more standardized compared to clinical chemistry measurements, it is essential to specify the instruments used for the Complete Blood Count (CBC). This is necessary to identify potential discrepancies in results (such as platelet counts) that may arise from different technologies. This factor is another key aspect affecting the reliability of tools developed using machine learning, as elucidated in [4].

4) Considering the three major flaws of the work, I prefer not to delve into the details of the manuscript. However, I would like to point out that there are also many small defects to be fixed, such as typographical and grammatical errors noted, indicating a need for careful proofreading. There are many acronyms that are not explained, or they are explained but not the first time they appear in the text. Figure 2 A does not include the units of measurement.

References

1. Campagner A, et al. External validation of Machine Learning models for COVID-19 detection based on Complete Blood Count. Health Inf Sci Syst. 2021 Oct 23;9(1):37. doi: 10.1007/s13755-021-00167-3.

2. Cabitza F, et al. The importance of being external. methodological insights for the external validation of machine learning models in medicine. Comput Methods Programs Biomed. 2021 Sep;208:106288. doi: 10.1016/j.cmpb.2021.106288.

3. Carobene A, et al. How is test laboratory data used and characterised by machine learning models? A systematic review of diagnostic and prognostic models developed for COVID-19 patients using only laboratory data. Clin Chem Lab Med. 2022 May 5;60(12):1887-1901. doi: 10.1515/cclm-2022-0182.

4. Campagner A, et al. Everything is varied: The surprising impact of instantial variation on ML reliability. Applied Soft Computing Volume 146 October 2023 Article number 110644 doi:10.1016/j.asoc.2023.110644

Reviewer #6: The study addresses the critical issue of COVID-19 diagnosis, leveraging machine learning (ML) techniques to analyze hematological parameters. Its novelty lies in assessing the external validity of these ML-based predictions across diverse populations from Brazil, Italy, and Western Europe, using a significant sample size of 195,554. However, as I will argue later, this is not an original idea and the authors missed to consider a relevant previous contribution t this matter.

But to begin, I want to point out a sentence that I found puzzling: the authors write that "For the first time, we showed that the models trained and tested on the same population produce significant accuracy." This statement is the most obvious thing that can be said in the context of validating machine learning models, and I think it is either a typo or a sentence that the authors should rephrase or delete (more importantly, what do they mean by "for the first time"?).

Second, I noticed that the authors did not consider perhaps the most important work conducting external validation of models that similarly try to diagnose covid-19 from routine blood chemistry data.

Cabitza, F., Campagner, A., Soares, F., de Guadiana-Romualdo, L. G., Challa, F., Sulejmani, A., ... & Carobene, A. (2021). The importance of being external. methodological insights for the external validation of machine learning models in medicine. Computer Methods and Programs in Biomedicine, 208, 106288.

(https://doi-org.unimib.idm.oclc.org/10.1016/j.cmpb.2021.106288)

The authors should therefore.

1) consider the degree to which their features overlap with the models presented in that work and, if there is any significant difference, justify that their model is more parsimonious or uses features that cost less to detect.

2) Consider whether their best model can outperform the one presented in that paper, which is already a few years old. If I am not mistaken, their model is also accessible with a free online application, accessible here: https://covid19bloodtests.pythonanywhere.com/.

3) In conducting step 3 they should plot their ROC curve and compare it with that of other authors. If their best model's curve intersects their best model's curve at some point, then they should not compare AUC scores to determine which is the best model. Instead, the comparison should be conducted by comparing the balanced accuracy, accuracy and F1 scores with the rspective confidence intervals.

4) I strongly recommend plotting even for their models (or at least their best model) an External Performance Diagram (accessible here: https://qualiml.pythonanywhere.com/?tool=pills-plt2). The similarity between (internal) and external training datasets can be calculated with this tool: https://psicorrespondence.pythonanywhere.com/. I am citing work that is all clearly stated in the work cited above.

I noticed that I was invited to review a second revision. I noticed that the reviewers involved in the first round raised three important issues, which I also share.

First, I also need a clarification on the handling of dataset imbalances and the rationale behind dataset variations. Second, I also believe that the manuscript needs a more robust justification for the practical applicability of their model. Utility, in terms of Standardized Net Benefit could be evaluated. Third, last but not the least, the previous reviewers noticed that the study did not provide a comprehensive comparison with existing literature. Since I also believe that the most important study to date on their same field has been neglected, I also believe that the authors should improve their work before this can be considered acceptable for publication for an important venue as Plos One. I would add that if the authors fail to demonstrate that their model is better than the one presented in Cabitza et al. for example, in balanced accuracy, F1 score, utility or robustness (as might be inferred from the External Performance diagram that I requested them to report in the manuscript) then their work should *not* be published, because it does not represent an innovation in the field compared to the available SOTA. In that cae, however, there might also be some value in testing the best SOTA with the datasets the authors have collected to check its practical robustness.

7. PLOS authors have the option to publish the peer review history of their article (what does this mean?). If published, this will include your full peer review and any attached files.

Reviewer #5: No

Reviewer #6: No

---

## [Author Response · Author response to Decision Letter 1]

31 Mar 2024

Response to the reviewers

Reviewer #5: The research is significant as it addresses the need for rapid, accurate, and cost-effective COVID-19 detection methods, especially in large-scale and low-income populations.

However, the manuscript, even in its revised version, still has numerous flaws that need to be addressed to allow for its publication.

1) The authors claim that used an innovative approach combining machine learning with hematological parameters for COVID-19 prediction, but they do not consider previous studies that have already been published on this matter as published in [1]. 

Ans: The above-mentioned reference (reference 35) was cited in the text (“..all-markups” file: line 105). The text was modified accordingly my removing the statements including innovative approaches (“..all-markups” file:lines 40-41)..

Again, authors claim that the novelty lies in the external validation of the ML models across different populations and countries, which has not been explored in previous studies. Unfortunately, they do not consider [2], a study published two years ago with different dataset collected across 3 different continents.

Ans: The above-mentioned reference (reference 36) was extensively cited in the text (“..all-markups” file: lines 105, 149, 350, 389, 609, 614, 618, 625, 632 and 644). The statement “the novelty lies in the external validation of the ML models across different populations and countries, which has not been explored in previous studies” was removed. 

2) The diminished effectiveness of the model when trained with data from one population and applied to another raises concerns about its wider applicability. 

Ans: “The diminished effectiveness of the model when trained with data from one population and applied to another” has been illustrated in detail, citing evidence from reference 36 and present results (“..all-markups” file:lines 144-149, lines 617-622, lines 629-638)

The authors mention that “the combination of CBC parameters varies with ethnicity” (line 98), yet they provide no further elaboration or references to support this assertion. It's crucial, in my view, for the authors to delve deeper into this premise, as it underpins the main argument against the reliability of tools utilizing laboratory data across diverse datasets. I highly recommend a thorough review of reference [3].

Ans: As suggested by the reviewer, we delved deeper into the point - “the combination of CBC parameters varies with ethnicity”. Elaboration and references are added in the text (“..all-markups” file:, lines 109-134). 

3) Another significant limitation of this paper is the complete absence of details about the methodologies employed for the laboratory tests. While it's acknowledged that hematological instruments tend to be more standardized compared to clinical chemistry measurements, it is essential to specify the instruments used for the Complete Blood Count (CBC). This is necessary to identify potential discrepancies in results (such as platelet counts) that may arise from different technologies. This factor is another key aspect affecting the reliability of tools developed using machine learning, as elucidated in [4].

Ans: As suggested by the reviewer, the information on the instruments (hematology analyzer) is mentioned in the text “..all-markups” file:line:192; lines 219-220; line:229-230. For datasets 1 and 3, those were not available. Only available for dataset 2.

4) Considering the three major flaws of the work, I prefer not to delve into the details of the manuscript. However, I would like to point out that there are also many small defects to be fixed, such as typographical and grammatical errors noted, indicating a need for careful proofreading. There are many acronyms that are not explained, or they are explained but not the first time they appear in the text. Figure 2 A does not include the units of measurement.

Ans: Units were not available for the dataset 1, based on which Figure 2A was generated

.

References

1. Campagner A, et al. External validation of Machine Learning models for COVID-19 detection based on Complete Blood Count. Health Inf Sci Syst. 2021 Oct 23;9(1):37. doi: 10.1007/s13755-021-00167-3.

2. Cabitza F, et al. The importance of being external. methodological insights for the external validation of machine learning models in medicine. Comput Methods Programs Biomed. 2021 Sep;208:106288. doi: 10.1016/j.cmpb.2021.106288.

3. Carobene A, et al. How is test laboratory data used and characterised by machine learning models? A systematic review of diagnostic and prognostic models developed for COVID-19 patients using only laboratory data. Clin Chem Lab Med. 2022 May 5;60(12):1887-1901. doi: 10.1515/cclm-2022-0182.

4. Campagner A, et al. Everything is varied: The surprising impact of instantial variation on ML reliability. Applied Soft Computing Volume 146 October 2023 Article number 110644 doi:10.1016/j.asoc.2023.110644

Reviewer #6: The study addresses the critical issue of COVID-19 diagnosis, leveraging machine learning (ML) techniques to analyze hematological parameters. Its novelty lies in assessing the external validity of these ML-based predictions across diverse populations from Brazil, Italy, and Western Europe, using a significant sample size of 195,554. However, as I will argue later, this is not an original idea and the authors missed to consider a relevant previous contribution t this matter.

But to begin, I want to point out a sentence that I found puzzling: the authors write that "For the first time, we showed that the models trained and tested on the same population produce significant accuracy." This statement is the most obvious thing that can be said in the context of validating machine learning models, and I think it is either a typo or a sentence that the authors should rephrase or delete (more importantly, what do they mean by "for the first time"?).

Ans: The statement has been removed and rephrased, as suggested by the reviewer (“..all-markups” file: lines 40-45)

Second, I noticed that the authors did not consider perhaps the most important work conducting external validation of models that similarly try to diagnose covid-19 from routine blood chemistry data.

Cabitza, F., Campagner, A., Soares, F., de Guadiana-Romualdo, L. G., Challa, F., Sulejmani, A., ... & Carobene, A. (2021). The importance of being external. methodological insights for the external validation of machine learning models in medicine. Computer Methods and Programs in Biomedicine, 208, 106288.

(https://doi-org.unimib.idm.oclc.org/10.1016/j.cmpb.2021.106288)

Ans: The above-mentioned reference (reference 36) was extensively cited in the text (“..all-markups” file: lines 105, 149, 350, 389, 609, 614, 618, 625, 632 and 644).

The authors should therefore.

1) consider the degree to which their features overlap with the models presented in that work and, if there is any significant difference, justify that their model is more parsimonious or uses features that cost less to detect.

Ans: Detail discussion was added (“..all-markups” file: lines 623-654)

2) Consider whether their best model can outperform the one presented in that paper, which is already a few years old. If I am not mistaken, their model is also accessible with a free online application, accessible here: https://covid19bloodtests.pythonanywhere.com/.

Ans: Yes, one of the models (trained on dataset from Italy and tested on dataset of Brazil) has outperformed the one presented in that paper (“..all-markups” file: line 642-646)

3) In conducting step 3 they should plot their ROC curve and compare it with that of other authors. If their best model's curve intersects their best model's curve at some point, then they should not compare AUC scores to determine which is the best model. Instead, the comparison should be conducted by comparing the balanced accuracy, accuracy and F1 scores with the rspective confidence intervals.

Ans: balanced accuracy, accuracy and F2 scores were reported (Table 3). 

4) I strongly recommend plotting even for their models (or at least their best model) an External Performance Diagram (accessible here: https://qualiml.pythonanywhere.com/?tool=pills-plt2). 

Ans: External performance diagram was plotted (Figure 6).

The similarity between (internal) and external training datasets can be calculated with this tool: https://psicorrespondence.pythonanywhere.com/. I am citing work that is all clearly stated in the work cited above.

Ans: The similarity between internal and external datasets were computed using Kolmogorov–Smirnov (KS) test (“..all-markups” file: line: 353-359). The online similarity tool indicated by the referee had some technical glitch; hence, we used the alternate method.

I noticed that I was invited to review a second revision. I noticed that the reviewers involved in the first round raised three important issues, which I also share.

First, I also need a clarification on the handling of dataset imbalances and the rationale behind dataset variations.

Ans: Imbalance in dataset was handled during external data validation by computing “Balanced Accuracy” and F2-score (“..all-markups” file: Table 3, line 360-380, 570, 592, 

Second, I also believe that the manuscript needs a more robust justification for the practical applicability of their model. Utility, in terms of Standardized Net Benefit could be evaluated. Third, last but not the least, the previous reviewers noticed that the study did not provide a comprehensive comparison with existing literature.

Ans: A new section has been added by comparing the results with the existing literature (“..all-markups” file: lines 623-654)

 Since I also believe that the most important study to date on their same field has been neglected, I also believe that the authors should improve their work before this can be considered acceptable for publication for an important venue as Plos One. I would add that if the authors fail to demonstrate that their model is better than the one presented in Cabitza et al. for example, in balanced accuracy, F1 score, utility or robustness (as might be inferred from the External Performance diagram that I requested them to report in the manuscript) then their work should *not* be published, because it does not represent an innovation in the field compared to the available SOTA. In that cae, however, there might also be some value in testing the best SOTA with the datasets the authors have collected to check its practical robustness.

Ans: I humbly ascertain that Balanced accuracy, F2 score were superior in some of the models presented in this study compared to the existing literature (we were unable to compute the standardized net benefit due to the technical glitch of the online tool to compute “External Performance Diagram”, suggested by the reviewer). SOTA is mentioned in the abstract and in the text (“..all-markups” file: lines 42-45 and 623-654)

---

## [Decision Letter · Decision Letter 2]

31 May 2024

PONE-D-23-15648R2The external validity of machine learning-based prediction scores from hematological parameters of COVID-19: A study using hospital records from Brazil, Italy, and Western EuropePLOS ONE

Dear Dr. Bandyopadhyay,

Thank you for submitting your manuscript to PLOS ONE. After careful consideration, we feel that it has merit but does not fully meet PLOS ONE’s publication criteria as it currently stands. Therefore, we invite you to submit a revised version of the manuscript that addresses the points raised during the review process. I have carefully evaluated the manuscript and the reviewers' comments. Below, I provide my evaluation and recommendations to the authors. First, the manuscript addresses a critical problem in the diagnosis of COVID-19 by exploiting machine learning techniques to analyze hematologic parameters. The novelty of the study is to evaluate the external validity of machine learning-based predictions in diverse populations, which is commendable. However, several problems and areas for improvement were identified by the reviewers that I agree with and pray for review. Although the manuscript addresses a major research gap, several corrections are needed to improve its clarity, rigor, and novelty. I recommend major revisions to comprehensively address the concerns raised by the reviewers. Once these revisions are implemented, we will evaluate whether the work meets PLOS ONE standards.

We look forward to receiving your revised manuscript.

Kind regards,

Francesco Branda, Ph.D.

Academic Editor

PLOS ONE

Additional Editor Comments:

Reviewers' comments:

Reviewer's Responses to Questions

**Comments to the Author**

1. If the authors have adequately addressed your comments raised in a previous round of review and you feel that this manuscript is now acceptable for publication, you may indicate that here to bypass the “Comments to the Author” section, enter your conflict of interest statement in the “Confidential to Editor” section, and submit your "Accept" recommendation.

Reviewer #5: (No Response)

Reviewer #6: (No Response)

2. Is the manuscript technically sound, and do the data support the conclusions?

Reviewer #5: Partly

Reviewer #6: Partly

3. Has the statistical analysis been performed appropriately and rigorously? 

Reviewer #5: Yes

Reviewer #6: Yes

4. Have the authors made all data underlying the findings in their manuscript fully available?

Reviewer #5: Yes

Reviewer #6: No

5. Is the manuscript presented in an intelligible fashion and written in standard English?

Reviewer #5: Yes

Reviewer #6: Yes

6. Review Comments to the Author

Reviewer #5: The manuscript has undoubtedly improved compared with the previous version. However, I am always surprised when authors only partially follow the suggestions of reviewers. On page 29, the authors stated:

"To note, machine learning (ML) models have inherent limitations, including dependency on datasets, size of the datasets, ethnic variabilities, phenotypic variabilities, and analytical instrumentation for clinical chemistry tests, etc."

Nevertheless, they failed to provide the necessary references as suggested in my previous revision. There is extensive literature on ML models developed using laboratory data that fails to acknowledge these data. As a result, most of these algorithms are not adopted in clinical practice. The specifics of the instrument, unit, and method are not "superfluous" information; without these details, the algorithm cannot be reproducibly applied in other clinical settings. This significant limitation of the study should be clearly stated.

Reviewer #6: I thank the authors for providing me with a point-by-point response letter. This is a very important thing that I also expect for the next response. I am happy to state that I believe the authors are on the right track. However, they have not yet reached a completely acceptable manuscript, for the reasons I'll share in what follows.

While I am very pleased that the authors have considered addressing robustness, I feel that my comment about the utility has been almost entirely disregarded. It is necessary for the authors to calculate the utility of the system, which is even more important since the authors insist on talking about it without measuring it ("The results promise the utility of these models when trained and tested on the same populations."). Personally, I am not aware of any other online tool for calculating the utility of a classification system other than the one available at this address: https://modelutility.pythonanywhere.com/. However, I think that even the tool used by the authors to calculate robustness should give an indication of the ECI or Brier Score ( which I find less understandable than the ECI). Also in regards to the AUC, I prefer the balanced accuracy score and I think the authors should report that in the robustness figure and not the AUC unless they convince me otherwise. Moreover, since the comparison with the State of the Art systems (SOTA) is critical, I am pleased that the authors show that one of their models is indeed better than the existing SOTA, but this noteworthy result should be shown in the robustness figure. SOTA has been shared on the Zenodo platform and is available online to test on authors' cases.

Since, as I already "disclosed" to the authors that I was enrolled as a reviewer during the second round to replace a reviewer who was no longer available, I want to make my own a comment that I feel was not adequately addressed by the authors regarding the absence of detailed information about the methodologies and instruments used for laboratory tests, particularly in the context of datasets and the corresponding units of measurement, In my opinion, the solution provided by the authors is still incomplete. However it is critical to underscore the importance of including comprehensive details on the instruments and units used across all datasets, not limited to a subset. The authors' reply addresses the instrument used only for dataset 2 and omits relevant information for datasets 1 and 3. This omission significantly impairs the scientific reliability and reproducibility of the results, which is necessary for any valuable research, but even crucial in studies involving machine learning algorithms. Indeed, the absence of such details can lead to inconsistent application and interpretation of the developed models, as laboratory measurements can vary significantly with different instruments and methodologies. Without standardization or at least a clear acknowledgment and documentation of the differences, any machine learning model developed from this data is at a risk of failing to generalize, thus limiting its practical utility and adoption in clinical settings. As noted in the literature, the lack of reproducibility and methodological transparency is a common reason for the limited real-world applicability of many research findings in the machine learning domain. In particular, the inability to verify the consistency of units and measurement techniques across datasets compromises the integrity of the analysis. I recommend that this limitation be clearly acknowledged in the manuscript to provide readers with a comprehensive understanding of the potential variability and implications for the study's outcomes.

Thus, all things considered, once authors will have addressed the above issue, as well as the utility issue, and improved the robustness diagram (which I really liked!) I believe the manuscript can be accepted for publication. Thank you for your comprehension and collaboration.

7. PLOS authors have the option to publish the peer review history of their article (what does this mean?). If published, this will include your full peer review and any attached files.

Reviewer #5: No

Reviewer #6: No

---

## [Author Response · Author response to Decision Letter 2]

13 Aug 2024

Review Comments to the Author

Reviewer #5: The manuscript has undoubtedly improved compared with the previous version. However, I am always surprised when authors only partially follow the suggestions of reviewers. On page 29, the authors stated:

"To note, machine learning (ML) models have inherent limitations, including dependency on datasets, size of the datasets, ethnic variabilities, phenotypic variabilities, and analytical instrumentation for clinical chemistry tests, etc."

Nevertheless, they failed to provide the necessary references as suggested in my previous revision. There is extensive literature on ML models developed using laboratory data that fails to acknowledge these data. As a result, most of these algorithms are not adopted in clinical practice. The specifics of the instrument, unit, and method are not "superfluous" information; without these details, the algorithm cannot be reproducibly applied in other clinical settings. This significant limitation of the study should be clearly stated.

Response: The reference is added on i) page 23; line 539 and page 30; line 712-714.

Reviewer #6: I thank the authors for providing me with a point-by-point response letter. This is a very important thing that I also expect for the next response. I am happy to state that I believe the authors are on the right track. However, they have not yet reached a completely acceptable manuscript, for the reasons I'll share in what follows.

While I am very pleased that the authors have considered addressing robustness, I feel that my comment about the utility has been almost entirely disregarded. It is necessary for the authors to calculate the utility of the system, which is even more important since the authors insist on talking about it without measuring it ("The results promise the utility of these models when trained and tested on the same populations.").

Response: The utility was estimated using standardized net benefit. The standardized net benefit equation was adopted from a cross-reference (equation 3 of Riley et.al. Statistics in Medicine, 2021, ug 30; 40(19):4230-51) of the reference, suggested by the reviewer; reference - 36 in this manuscript. As the online tool, suggested by the reviewer, is currently not working we could not replot the robustness diagram. The standardized net benefit values at different threshold values were reported in Table 3. 

 Personally, I am not aware of any other online tool for calculating the utility of a classification system other than the one available at this address: https://modelutility.pythonanywhere.com/. However, I think that even the tool used by the authors to calculate robustness should give an indication of the ECI or Brier Score ( which I find less understandable than the ECI). Also in regards to the AUC, I prefer the balanced accuracy score and I think the authors should report that in the robustness figure and not the AUC unless they convince me otherwise. 

Response: AUC and Brier scores were depicted against the similarity values in external performance diagram (Figure 6 and page 27; line 636). Although, the reviewer has suggested to report balanced accuracy score instead of AUC in the robustness figure, the online tool (https://modelutility.pythonanywhere.com) only provides AUC score and not the balanced accuracy, in the figure. To note, the tool is currently (as of on 8th August 2024) not working. Hence, we could not replot the figure. The advantage of balanced accuracy over AUC score was discussed (page 23; line 551-553). It is worthy to mention that we have not used any online tools. Instead, we used the equations given in several references and have written codes on our own to deduce the values.

Moreover, since the comparison with the State of the Art systems (SOTA) is critical, I am pleased that the authors show that one of their models is indeed better than the existing SOTA, but this noteworthy result should be shown in the robustness figure. SOTA has been shared on the Zenodo platform and is available online to test on authors' cases.

Response: We cannot show the robustness figure as the online tool suggested by the reviewer was not working, currently. However, we tried our best to consolidate the results of utility (in terms of standardized net benefit) in Table 3 and in the text – (page 16: line 361 – line 368); (page 23: line 553 – page 24: line 562); (page 25:line 584-page 26:line 591); (page 27:642 – page 28:651)

Since, as I already "disclosed" to the authors that I was enrolled as a reviewer during the second round to replace a reviewer who was no longer available, I want to make my own a comment that I feel was not adequately addressed by the authors regarding the absence of detailed information about the methodologies and instruments used for laboratory tests, particularly in the context of datasets and the corresponding units of measurement, In my opinion, the solution provided by the authors is still incomplete. However it is critical to underscore the importance of including comprehensive details on the instruments and units used across all datasets, not limited to a subset. The authors' reply addresses the instrument used only for dataset 2 and omits relevant information for datasets 1 and 3. This omission significantly impairs the scientific reliability and reproducibility of the results, which is necessary for any valuable research, but even crucial in studies involving machine learning algorithms. Indeed, the absence of such details can lead to inconsistent application and interpretation of the developed models, as laboratory measurements can vary significantly with different instruments and methodologies. Without standardization or at least a clear acknowledgment and documentation of the differences, any machine learning model developed from this data is at a risk of failing to generalize, thus limiting its practical utility and adoption in clinical settings. As noted in the literature, the lack of reproducibility and methodological transparency is a common reason for the limited real-world applicability of many research findings in the machine learning domain. In particular, the inability to verify the consistency of units and measurement techniques across datasets compromises the integrity of the analysis. I recommend that this limitation be clearly acknowledged in the manuscript to provide readers with a comprehensive understanding of the potential variability and implications for the study's outcomes.

Response: The limitation was clearly acknowledged in page 29; line 712-714, with proper reference. The reference was also added to page 23; line 539.

Thus, all things considered, once authors will have addressed the above issue, as well as the utility issue, and improved the robustness diagram (which I really liked!) I believe the manuscript can be accepted for publication. Thank you for your comprehension and collaboration.

---

## [Decision Letter · Decision Letter 3]

21 Aug 2024

PONE-D-23-15648R3The external validity of machine learning-based prediction scores from hematological parameters of COVID-19: A study using hospital records from Brazil, Italy, and Western EuropePLOS ONE

Dear Dr. Bandyopadhyay,

Thank you for submitting your manuscript to PLOS ONE. After careful consideration, we feel that it has merit but does not fully meet PLOS ONE’s publication criteria as it currently stands. Therefore, we invite you to submit a revised version of the manuscript that addresses the points raised during the review process.

Although significant improvements have been made, however, there are still some things to be adjusted as suggested in the last review. Please proceed with what has been requested. 

We look forward to receiving your revised manuscript.

Kind regards,

Francesco Branda, Ph.D.

Academic Editor

PLOS ONE

Reviewers' comments:

Reviewer's Responses to Questions

**Comments to the Author**

1. If the authors have adequately addressed your comments raised in a previous round of review and you feel that this manuscript is now acceptable for publication, you may indicate that here to bypass the “Comments to the Author” section, enter your conflict of interest statement in the “Confidential to Editor” section, and submit your "Accept" recommendation.

Reviewer #5: All comments have been addressed

Reviewer #6: (No Response)

2. Is the manuscript technically sound, and do the data support the conclusions?

Reviewer #5: Yes

Reviewer #6: Yes

3. Has the statistical analysis been performed appropriately and rigorously? 

Reviewer #5: Yes

Reviewer #6: No

4. Have the authors made all data underlying the findings in their manuscript fully available?

Reviewer #5: Yes

Reviewer #6: No

5. Is the manuscript presented in an intelligible fashion and written in standard English?

Reviewer #5: Yes

Reviewer #6: Yes

6. Review Comments to the Author

Reviewer #5: The authors have made considerable efforts to revise the manuscript, and I acknowledge the improvements that have been made. However, I remain concerned about the lack of novelty in this work, which was a primary point of feedback in earlier reviews. The extensive literature in this field has already covered much of what is presented here, and this diminishes the impact of the current study.

Regarding the suggestion to include the review, the intention was not merely to add a citation but to encourage the authors to consider the issue of "noise" in laboratory data—a factor often overlooked, which leads to the poor performance and limited adoption of these tools in practice. This aspect remains unaddressed.

The manuscript would have benefited from a more in-depth exploration of the "noise" in laboratory data (see https://doi.org/10.1016/j.asoc.2023.110644). Including such considerations would have added a layer of depth and relevance to the study, making it more compelling.

While the paper may be suitable for publication, it is unfortunate that, despite the detailed point-by-point revisions received, the authors have missed the opportunity to make it truly unique and of significant novelty in the field.

Reviewer #6: The article is much improved, I must give credit to the authors and indeed thank them for considering my suggestions. In my opinion now only the robustness diagram is missing, to compare it with the previous SOTA (which is the article 36). I personally checked that my previous suggestion was not feasible, as written by the authors in their response, but when I checked all the reported tools were working; to be precise, the tool for evaluating model calibration is available at https://calibrationassessment.pythonanywhere.com/; the one for the utility is available here: https://modelutility.pythonanywhere.com/; the one for calculating similarity here: https://psicorrespondence.pythonanywhere.com/ and finally the one for the robustness diagram here: https://qualiml.pythonanywhere.com/. I have no special contact with the authors of those tools: I just looked in the html code of the page I had reported earlier, namely: https://mudilab.github.io/dss-quality-assessment/ and extracted the links to the individual tools.

I believe that adding and discussing that diagram and making the codes and datasets available are necessary and sufficient steps to make the work of adequate quality for publication.

7. PLOS authors have the option to publish the peer review history of their article (what does this mean?). If published, this will include your full peer review and any attached files.

Reviewer #5: No

Reviewer #6: No

---

## [Author Response · Author response to Decision Letter 3]

5 Oct 2024

Review Comments to the Author

Reviewer #5: The authors have made considerable efforts to revise the manuscript, and I acknowledge the improvements that have been made. However, I remain concerned about the lack of novelty in this work, which was a primary point of feedback in earlier reviews. The extensive literature in this field has already covered much of what is presented here, and this diminishes the impact of the current study.

Regarding the suggestion to include the review, the intention was not merely to add a citation but to encourage the authors to consider the issue of "noise" in laboratory data—a factor often overlooked, which leads to the poor performance and limited adoption of these tools in practice. This aspect remains unaddressed.

The manuscript would have benefited from a more in-depth exploration of the "noise" in laboratory data (see https://doi.org/10.1016/j.asoc.2023.110644). Including such considerations would have added a layer of depth and relevance to the study, making it more compelling.

While the paper may be suitable for publication, it is unfortunate that, despite the detailed point-by-point revisions received, the authors have missed the opportunity to make it truly unique and of significant novelty in the field.

Response: The reviewer’s comment has been addressed by adding a separate section on “Impact of Instantial Variation (IV) on the external validation datasets:” pages: 27-29; lines: 660-704. Also mentioned in the conclusion, lines: 755-757 (line numbers according to version marked with markup)

Reviewer #6: The article is much improved, I must give credit to the authors and indeed thank them for considering my suggestions. In my opinion now only the robustness diagram is missing, to compare it with the previous SOTA (which is the article 36). I personally checked that my previous suggestion was not feasible, as written by the authors in their response, but when I checked all the reported tools were working; to be precise, the tool for evaluating model calibration is available at https://calibrationassessment.pythonanywhere.com/; the one for the utility is available here: https://modelutility.pythonanywhere.com/; the one for calculating similarity here: https://psicorrespondence.pythonanywhere.com/ and finally the one for the robustness diagram here: https://qualiml.pythonanywhere.com/. I have no special contact with the authors of those tools: I just looked in the html code of the page I had reported earlier, namely: https://mudilab.github.io/dss-quality-assessment/ and extracted the links to the individual tools.

I believe that adding and discussing that diagram and making the codes and datasets available are necessary and sufficient steps to make the work of adequate quality for publication.

Response: Finally, we were able to execute the online tool with a different browser. Robustness diagram is now added (Figure 6); lines: 599, 605.

---

## [Decision Letter · Decision Letter 4]

5 Nov 2024

PONE-D-23-15648R4The external validity of machine learning-based prediction scores from hematological parameters of COVID-19: A study using hospital records from Brazil, Italy, and Western EuropePLOS ONE

Dear Dr. Bandyopadhyay,

Thank you for submitting your manuscript to PLOS ONE. After careful consideration, we feel that it has merit but does not fully meet PLOS ONE’s publication criteria as it currently stands. Therefore, we invite you to submit a revised version of the manuscript that addresses the points raised during the review process.

I recommend resolving the last issue raised by the reviewer. 

We look forward to receiving your revised manuscript.

Kind regards,

Francesco Branda, Ph.D.

Academic Editor

PLOS ONE

Journal Requirements:

Reviewers' comments:

Reviewer's Responses to Questions

**Comments to the Author**

1. If the authors have adequately addressed your comments raised in a previous round of review and you feel that this manuscript is now acceptable for publication, you may indicate that here to bypass the “Comments to the Author” section, enter your conflict of interest statement in the “Confidential to Editor” section, and submit your "Accept" recommendation.

Reviewer #5: All comments have been addressed

Reviewer #6: All comments have been addressed

2. Is the manuscript technically sound, and do the data support the conclusions?

Reviewer #5: Yes

Reviewer #6: Yes

3. Has the statistical analysis been performed appropriately and rigorously? 

Reviewer #5: Yes

Reviewer #6: Yes

4. Have the authors made all data underlying the findings in their manuscript fully available?

Reviewer #5: Yes

Reviewer #6: Yes

5. Is the manuscript presented in an intelligible fashion and written in standard English?

Reviewer #5: Yes

Reviewer #6: Yes

6. Review Comments to the Author

Reviewer #5: I would like to thank the authors for addressing all comments and suggestions thoroughly. The paper is truly of high quality—congratulations!

Reviewer #6: Authors addressed all my requests. However, to enable reproducibility and acknowledge the due credit, they should state, in the caption on Figure 6, the URL (web address) of the tool they used to generate the Robustness Diagram. Once this has been done, the manuscript can be accepted without further review.

7. PLOS authors have the option to publish the peer review history of their article (what does this mean?). If published, this will include your full peer review and any attached files.

Reviewer #5: No

Reviewer #6: No

---

## [Author Response · Author response to Decision Letter 4]

7 Nov 2024

Reviewer #6: Authors addressed all my requests. However, to enable reproducibility and acknowledge the due credit, they should state, in the caption on Figure 6, the URL (web address) of the tool they used to generate the Robustness Diagram.

Response: The URL (web address) has been added to the caption on Figure 6

---

## [Decision Letter · Decision Letter 5]

12 Dec 2024

The external validity of machine learning-based prediction scores from hematological parameters of COVID-19: A study using hospital records from Brazil, Italy, and Western Europe

PONE-D-23-15648R5

Dear Dr. Bandyopadhyay,

We’re pleased to inform you that your manuscript has been judged scientifically suitable for publication and will be formally accepted for publication once it meets all outstanding technical requirements.

Kind regards,

Francesco Branda, Ph.D.

Academic Editor

PLOS ONE

Additional Editor Comments (optional):

Reviewers' comments:

Reviewer's Responses to Questions

**Comments to the Author**

1. If the authors have adequately addressed your comments raised in a previous round of review and you feel that this manuscript is now acceptable for publication, you may indicate that here to bypass the “Comments to the Author” section, enter your conflict of interest statement in the “Confidential to Editor” section, and submit your "Accept" recommendation.

Reviewer #6: All comments have been addressed

2. Is the manuscript technically sound, and do the data support the conclusions?

Reviewer #6: Yes

3. Has the statistical analysis been performed appropriately and rigorously? 

Reviewer #6: Yes

4. Have the authors made all data underlying the findings in their manuscript fully available?

Reviewer #6: Yes

5. Is the manuscript presented in an intelligible fashion and written in standard English?

Reviewer #6: Yes

6. Review Comments to the Author

Reviewer #6: The revised version can be accepted for publication. As I said in my previous review a further review was not necessary: I requested only a small revision to do.

7. PLOS authors have the option to publish the peer review history of their article (what does this mean?). If published, this will include your full peer review and any attached files.

Reviewer #6: No

---

## [Editor Report · Acceptance letter]

20 Dec 2024

PONE-D-23-15648R5 

PLOS ONE

Dear Dr. Bandyopadhyay, 

I'm pleased to inform you that your manuscript has been deemed suitable for publication in PLOS ONE. Congratulations! Your manuscript is now being handed over to our production team.

Kind regards, 

on behalf of

Dr. Francesco Branda 

Academic Editor

PLOS ONE